# Emergent interface vibrational structure of oxide superlattices

Eric R. Hoglund[1✉], De-Liang Bao[2], Andrew O'Hara[2], Sara Makarem[1], Zachary T. Piontkowski[3], Joseph R. Matson[4], Ajay K. Yadav[5], Ryan C. Haislmaier[6], Roman Engel-Herbert[7,8], Jon F. Ihlefeld[1], Jayakanth Ravichandran[9], Ramamoorthy Ramesh[5], Joshua D. Caldwell[4], Thomas E. Beechem[3,10,11], John A. Tomko[12], Jordan A. Hachtel[13✉], Sokrates T. Pantelides[2,14✉], Patrick E. Hopkins[1,12,15✉] & James M. Howe[1✉]

As the length scales of materials decrease, the heterogeneities associated with interfaces become almost as important as the surrounding materials. This has led to extensive studies of emergent electronic and magnetic interface properties in superlattices[1–9]. However, the interfacial vibrations that affect the phonon-mediated properties, such as thermal conductivity[10,11], are measured using macroscopic techniques that lack spatial resolution. Although it is accepted that intrinsic phonons change near boundaries[12,13], the physical mechanisms and length scales through which interfacial effects influence materials remain unclear. Here we demonstrate the localized vibrational response of interfaces in strontium titanate–calcium titanate superlattices by combining advanced scanning transmission electron microscopy imaging and spectroscopy, density functional theory calculations and ultrafast optical spectroscopy. Structurally diffuse interfaces that bridge the bounding materials are observed and this local structure creates phonon modes that determine the global response of the superlattice once the spacing of the interfaces approaches the phonon spatial extent. Our results provide direct visualization of the progression of the local atomic structure and interface vibrations as they come to determine the vibrational response of an entire superlattice. Direct observation of such local atomic and vibrational phenomena demonstrates that their spatial extent needs to be quantified to understand macroscopic behaviour. Tailoring interfaces, and knowing their local vibrational response, provides a means of pursuing designer solids with emergent infrared and thermal responses.

The hierarchy of lattices in superlattices presents a tunable phonon–material interaction where, at small-to-moderate-period thicknesses, coherent and localized interface phonons have a major role in controlling properties. The vibrations and coupling present at interfaces in superlattices, in a broader context, occur at other interphase and intergranular boundaries and can result in remarkable properties[3,14–24]. Probing vibrations with the lateral spatial resolution required to provide knowledge that can be used for interface engineering and customization of thermal and infrared properties has remained prohibitively difficult[10,12,15,25–27]. The spectral and spatial resolution of monochromated electron energy loss spectroscopy (EELS) in a scanning transmission electron microscope (STEM) provides a unique opportunity to probe the spatial extent of vibrational excitations that are conventionally assessed by infrared light or neutrons. Such resolving capabilities have so far been demonstrated in resolving phonons associated with chemical changes at point defects and stacking faults in crystals[28–31].

Segmented STEM detectors used for integrated differential phase contrast (iDPC) can image both light and heavy elements, providing knowledge of local symmetry, which dictates vibrational properties[32–34]. Hence, STEM imaging and EELS provides a toolset to understand the intertwined local symmetry and vibrational properties at material interfaces.

Here we combine advanced STEM-iDPC and monochromated EELS experiments with density functional theory (DFT) calculations to

[1]Department of Materials Science and Engineering, University of Virginia, Charlottesville, VA, USA. [2]Department of Physics and Astronomy, Vanderbilt University, Nashville, TN, USA. [3]Sandia National Laboratories, Albuquerque, NM, USA. [4]Department of Mechanical Engineering and Electrical Engineering, Vanderbilt University, Nashville, TN, USA. [5]Department of Materials Science and Engineering, University of California Berkley, Berkley, CA, USA. [6]Department of Materials Science and Engineering, Pennsylvania State University, University Park, PA, USA. [7]Paul-Drude-Institut für Festkörperelektronik, Berlin, Germany. [8]Institut für Physik, Humboldt-Universität zu Berlin, Berlin, Germany. [9]Department of Chemical Engineering and Materials Science, University of Southern California, Los Angeles, CA, USA. [10]Center for Integrated Nanotechnologies, Sandia National Laboratories, Albuquerque, NM, USA. [11]School of Mechanical Engineering and the Birck Nanotechnology Center, Purdue University, West Lafayette, IN, USA. [12]Department of Mechanical and Aerospace Engineering, University of Virginia, Charlottesville, VA, USA. [13]Center for Nanophase Materials Sciences, Oak Ridge National Laboratory, Oak Ridge, TN, USA. [14]Department of Electrical and Computer Engineering, Vanderbilt University, Nashville, TN, USA. [15]Department of Physics, University of Virginia, Charlottesville, VA, USA. ✉e-mail: erh3cq@virginia.edu; hachtelja@ornl.gov; pantelides@vanderbilt.edu; phopkins@virginia.edu; jh9s@virginia.edu

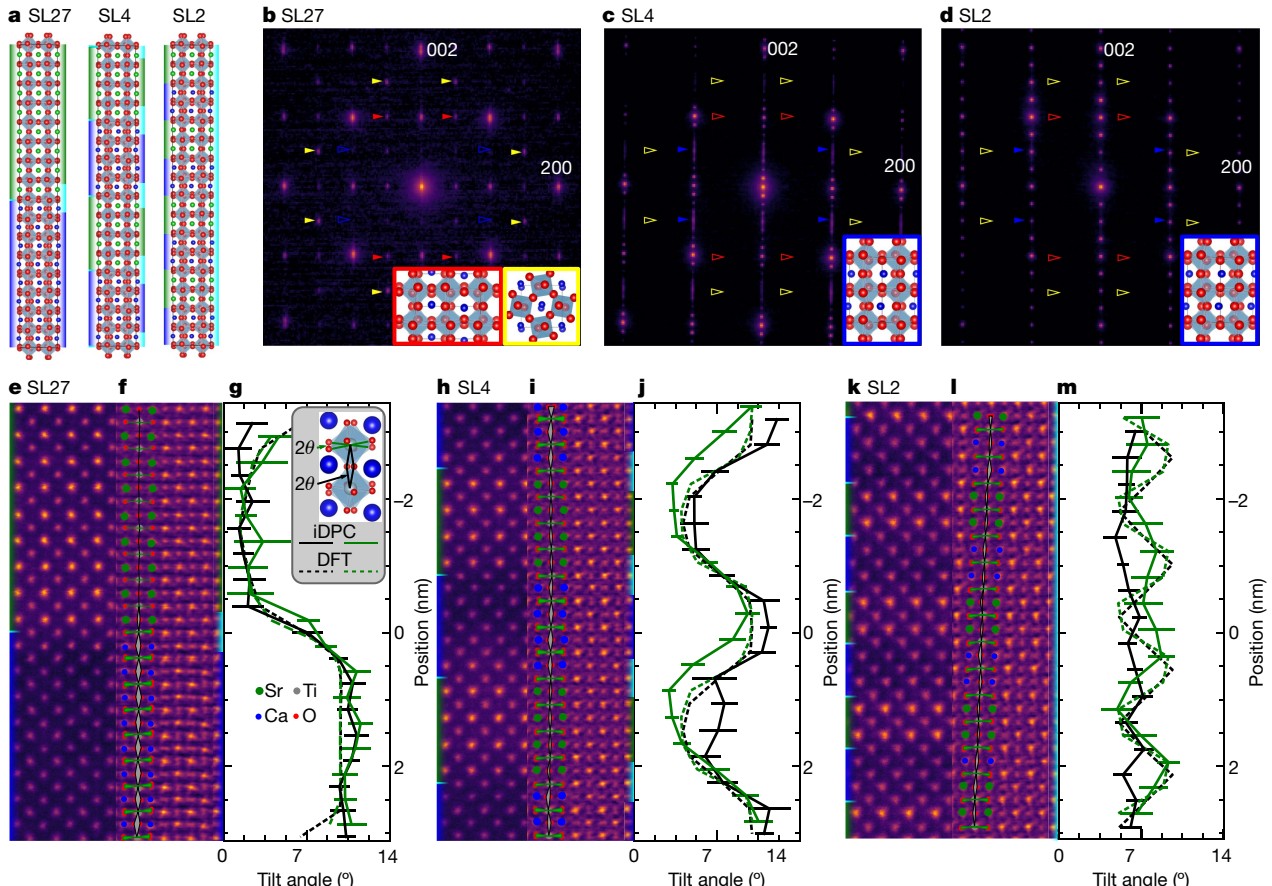

**Fig. 1 | Period-dependent changes in the symmetry of STO–CTO superlattices. a**, Superlattice structures calculated from DFT with coloured-bar schematics denoting the chemically (left) and structurally (right) defined interfaces. Here green, blue and cyan rectangles correspond to STO, CTO and interface layers, respectively; the same colours are used in **e**, **f**, **h**, **i**, **k**, **l**. Green, blue, grey and red circles in **a**, **e**, **f**, **h**, **i**, **k** correspond to Sr, Ca, Ti and O atoms, respectively. **b**–**d**, The [100] zone-axis SADP for SL27 (**b**), SL4 (**c**) and SL2 (**d**) grown on NGO. The coloured arrows correspond to ordered reflections from the three possible domains. The solid arrows indicate ordered reflections that exist and the hollow arrows indicate absences. Insets: ball-and-stick models of the orientations present with border colours matching the arrows. The red and blue arrows and insets are viewed along an out-of-phase tilt axis and the yellow are viewed along an in-phase tilt axis. In **c**, **d**, superlattice reflections are seen in the 001 direction. In **b**, closely spaced superlattice

reflections appear as streaking of the fundamental reflections. **e**–**m**, ADF images (**e**, **h**, **k**), iDPC images (**f**, **i**, **l**) and octahedral tilt angles (**g**, **j**, **m**) of SL27 (**e**–**g**), SL4 (**h**–**j**) and SL2 (**k**–**m**). The legend in **g** illustrates the in-plane (green) and out-of-plane (black) tilt angles (θ), which are defined as half of the projected O–Ti–O bond angle . The tilt angles for a one unit-cell column are overlayed in each iDPC image to demonstrate the changing in-plane (green triangles) and out-of-plane (grey triangles) tilt angles. In **g**, **j**, **m**, solid and dashed curves are from experimental measurements and calculations, respectively. The error bars represent one standard deviation. Chemically abrupt interfaces are illustrated to the left of the ADF images (**e**, **h**, **k**) and model structures (**a**), illustrating the abrupt change between STO (green) and CTO (blue) layers. Chemically diffuse interfaces are illustrated to the right of the iDPC images (**f**, **i**, **l**) and model structures (**a**), illustrating the non-abrupt symmetry changes that are occurring as a result of octahedral coupling.

quantify the local symmetry and vibrational states in strontium titanate (SrTiO$_3$, STO)–calcium titanate (CaTiO$_3$, CTO) superlattices. We measure the spatial extent of TiO$_6$ octahedral rotation across the STO–CTO interfaces (that is, octahedral coupling) and relate this information to the local titanium (Ti) and oxygen (O) vibrational response measured with high-spatial-resolution EELS. Second-harmonic-generation (SHG) measurements are performed to measure the macroscopic opto-electronic properties and interface density. DFT calculations are used to model the structural evolution of the superlattices and provide insights into the origins of their differing vibrational states. Finally, through ultrafast optical spectroscopy measurements, we assess the lifetime of the zone-centre phonon modes, providing insight into the macroscopic property progression observed in these oxide heterostructures. We show that as the superlattice layer thickness approaches the width of the structurally diffuse interfaces, where octahedral coupling occurs, the layers lose uniqueness and adopt the structure and vibrational response of the interface. Thus, the vibrational response of the interface becomes characteristic of the entire material.

To evaluate the influence of interfaces, we synthesized five STO–CTO superlattices with layer thicknesses of 27, 6, 4, 3 and 2 pseudo-cubic unit cells (SL27, SL6, SL4, SL3 and SL2, respectively). For (S)TEM and experiments we focus on SL27, SL4 and SL2, which are shown schematically in Fig. 1a. Large-period SL27 and short-period SL2 were chosen to represent superlattices with well separated and closely spaced interfaces, respectively. SL4 was chosen as an intermediate. SL6 and SL3 were included for the optics-based experiments to provide additional data points.

To quantify the structure, we acquired selected-area electron diffraction patterns (SADPs). The SADPs reveal that the orientation of octahedral tilts are different in the SL27 structure compared with the SL4 and SL2 structures. The large-period SL27 structure shows ordered reflections from two of three possible *Pbnm*-CTO domains (Fig. 1b), each with an in-plane *c* axis as illustrated by the ball-and-stick insets. Ordered reflections observed from the SL4 and SL2 samples indicate a single out-of-plane *c* axis, as shown in Fig. 1c, d and simulated in Supplementary Figs. 1, 2. This microstructural transition is accompanied by a relaxation of the lattice parameters to a single intermediate value

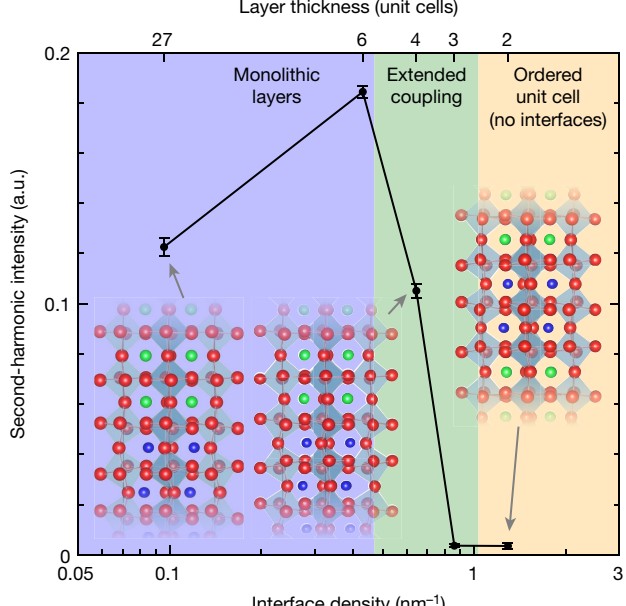

**Fig. 2 | Second-harmonic intensity indicates short-period superlattices lack interfaces.** Second-harmonic intensity of STO–CTO superlattices with varying periodicity, demonstrating various regimes of structural transitions and their role in the electronic/optical properties of heterostructures. The error bars are calculated from the mean square deviation of a parabolic fit to the measured second-harmonic intensity versus incident electric field. Ball-and-stick models are included to pictorially show the connection between octahedral tilt and the presence structurally diffuse interfaces, or lack thereof.

(Supplementary Fig. 3, Supplementary Tables 1, 2). Thus, as the layer thickness decreases, the underlying crystal structure adapts, which could be enabled by octahedral tilt[3,4,19,21,22].

To investigate the potential of octahedral tilting in the superlattices further, we use annular dark-field (ADF) and iDPC imaging to quantify the octahedral tilting (Fig. 1e–m). The atomic-number ($Z$) contrast of the ADF images (Fig. 1e, h, k) allows for discrimination between the brighter strontium (Sr) and darker calcium (Ca) atoms and shows chemically abrupt transitions between the two. In-plane (green triangles) and out-of-plane (grey triangles) tilt angles are overlaid on each iDPC image (Fig. 1f, i, l) to demonstrate the changing octahedral tilt angles from layer to layer and from structure to structure. For example, the iDPC image of SL27 (Fig. 1f) has three regions defined by observing the splitting of oxygen columns: single columns in STO, split columns in CTO and an intermediate splitting at the interface. The tilts are consistent with simple cubic $Pm\bar{3}m$-STO containing no tilt, orthorhombic $Pbnm$-CTO along an out-of-phase tilt axis and a region where tilts transition from finite angles in CTO to none in STO. Scanning convergent-beam electron diffraction corroborates these observations (Supplementary Fig. 4).

To quantify the changes in crystal structure with reduced interface separation, the in-plane and out-of-plane octahedral angles are measured (see the legend in Fig. 1g). The plane averaged tilt angle is shown in Fig. 1g, j, m. A coupling region is present at the STO–CTO interface of SL27 (Fig. 1g). Similar coupling regions observed in other perovskite heterostructures result in extraordinary electrical and magnetic properties[3,7–9,19–22]. We define the structurally diffuse interface width as one unit cell centred at the chemically abrupt TiO₂ interface as schematically shown in Fig. 1a.

We then turn to SL4 and SL2 to understand how the octahedra couple across the interfaces when the interface spacing is comparable to the diffuse interface width. The CTO oxygen column splitting in SL4 is less pronounced than in SL27. Some oxygen columns within the STO layers are distinctly split whereas most appear elliptical from the

partial overlap of the splitting column. The octahedra in STO and CTO are coupled, as seen in the sinusoidal profile of the tilt angles (Fig. 1j). Coupling is even more apparent in SL2 (Fig. 1m), where a nearly constant tilt angle of 7° extends throughout the entire structure. Similar titling of STO octahedra has also been observed in short-period barium titanium oxide (BaTiO₃)–STO superlattices[15]. Here we show that incorporation of the atomic displacements in STO is an interface-mediated process.

Using the STEM results, we define three types of superlattice: long-period superlattices, such as SL27, which exhibit monolithic phases with structurally diffuse interfaces; moderate-period superlattices (SL4) with modified monolithic phases and structurally diffuse interfaces taking up a sizable fraction of the superlattice; and short-period superlattices (SL2) comprising entirely interface regions that are better described as an ordered structure, with a global symmetry characteristic of that seen at the interfaces of all superlattices. In SL2, the material is more accurately described as having chemically ordered unit cells with a single tilt angle and a $Sr_2Ca_2Ti_4O_{12}$ basis[5,35,36]. More simply, the short-period superlattice has become a crystal of interfaces.

We perform SHG measurements to provide further evidence of the structural regimes, which also lends insight into the electronic properties of the superlattices, as shown in Fig. 2. Both STO and CTO have inversion symmetry, so the SHG intensity of STO–CTO superlattices is directly related to the polarizability of the interface where inversion symmetry is broken. An increase in SHG intensity is observed from SL27 to SL6 as the density of interfaces increases and layers remain independent with respect to each other. However, a marked decrease in the nonlinear optical response is observed in SL4 and rapidly vanishes in superlattices with short periodicities; as the heterostructure transitions from independent monolithic layers, to coupled layers, to a single centrosymmetric structure, the second-order optical response approaches zero. In other words, the structural transitions are directly reflected in the polarizability of the superlattices.

To support the observed crystallographic structure of the super-lattices and predict the effect of the observed local symmetries on vibrations, we performed DFT calculations on several prototypical superlattice models (Fig. 1a) and further consider metastable phases and layer intermixing in Supplementary Fig. 5. Octahedral tilt angles for both the experimental and theoretical results are in good agreement in both amplitude and periodicity, as indicated by the dashed curves shown in Fig. 1g, j, m. From both the structural calculations and experimental measurements, we can conclude that as the period thickness decreases, the system converges toward a single, emergent structure.

As phonon frequencies are affected by changes of bond lengths and bond angles, we expect the evolution of the octahedral tilts in our superlattices (Fig. 1) to affect the phonon density of states (PDOS), which drives the inherent thermal and infrared optical properties. To evaluate this possibility, we employed DFT to calculate the PDOS projected on the O and Ti atoms for each of the three superlattices characterized by STEM (Fig. 3a, Methods). The PDOS show three peaks at about 37 meV, about 60 meV and about 97 meV, which are further discussed in the Supplementary Information. From SL27 to SL4 and SL2, the 37-meV and 60-meV peaks redshift, whereas the 97-meV peak blueshifts, indicating an evolution of the superlattice phonon modes as the layer thickness decreases. Thus, vibrational modes assigned to the octahedra change energy as the octahedra they derive from change tilt angle with decreasing superlattice period thickness.

The attribution of peak shifts to changes in octahedral tilts is further supported by comparing the PDOS for each constituent layer and for the interface in the three superlattices (Fig. 3b). We see that in the SL27, the total DOS for the system deviates from the STO, CTO and interface spectra. However, in SL4 and SL2 the total DOS tracks the interface spectrum almost perfectly, illustrating how the interface dominates these shorter-period structures. Supplementary Figs. 6, 8b, c show examples of layer and interface modes. In other words, as the layer

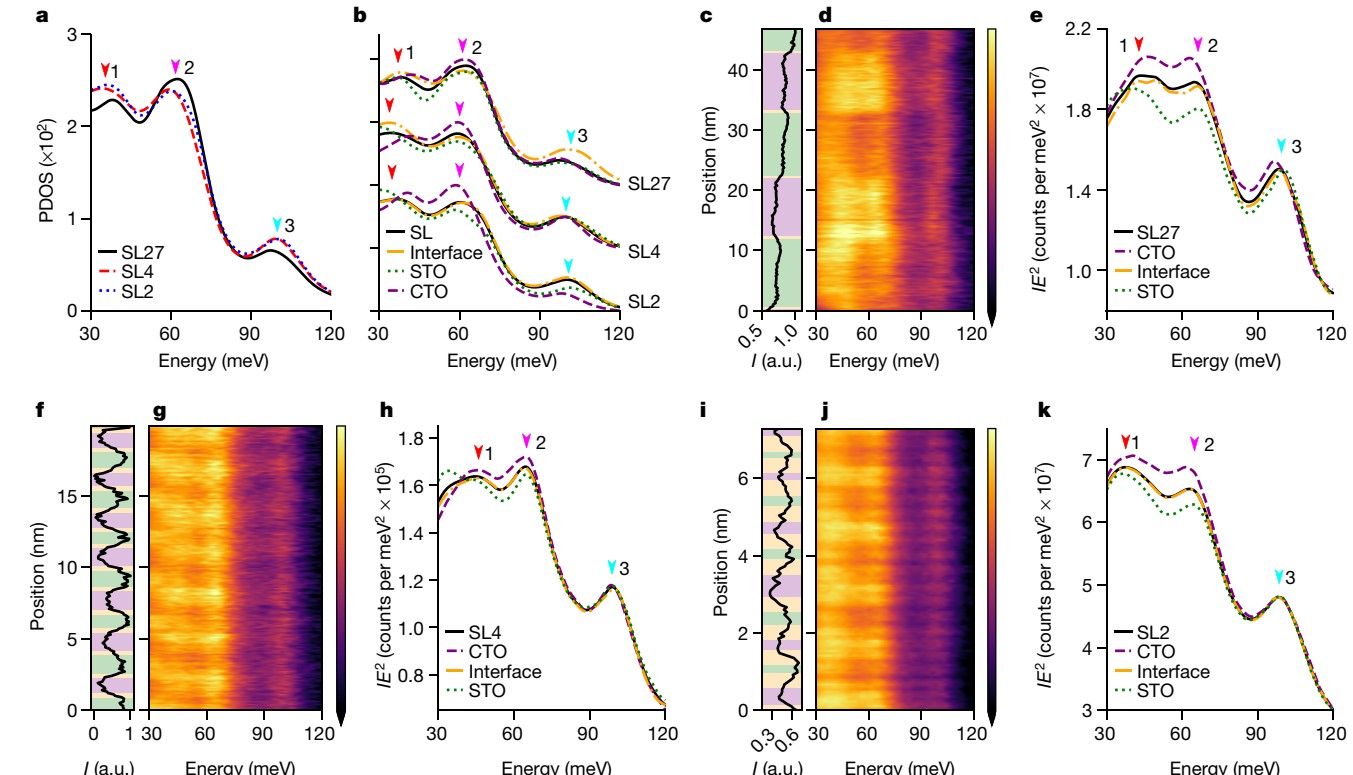

**Fig. 3 | Localized vibrational response of superlattices indicates the emergent role of the interfacial symmetry. a**, DFT-calculated PDOS projected on the octahedron O and Ti atoms of the SL27, SL4 and SL2 models. The arrows indicate the dominant phonon peaks. **b**, Cascade of DFT-calculated PDOS projected on STO (green), CTO (purple) and interface (orange) layers and the total DOS (black) for each superlattice model. **c**–**k**, Monochromated

STEM-EELS line profile analyses of the three SL structures SL27 (**c**–**e**), SL4 (**f**–**h**) and SL2 (**i**–**k**) with the ADF intensity (*I*) profile (**c**, **f**, **i**), EELS profile (**d**, **g**, **j**) and integrated spectra from each layer (**e**, **h**, **k**) (as indicated by coloured regions in the ADF profile). Energy-loss spectra are normalized by multiplying intensity by the energy squared ($IE^2$). The colour bars in **d**, **g**, **j** share the same labels and scales as **e**, **h**, **k**.

thickness decreases, both the structural and vibrational state converge towards the respective state of the interfaces.

The evolution in vibrational response of the superlattice is observed experimentally via spatially resolved off-axis vibrational EELS[29,31]. With this approach, the difference in vibrational response within the diffuse interface region can be directly compared with that within the constituent layers themselves. The ADF line profiles of SL27, SL4 and SL2 are shown in Fig. 3c, f, i, and distinguish between the heavy STO and light CTO layers. The simultaneously acquired EELS are shown in Fig. 3d, g, j. Changes in the layer-to-layer response are clearly observed in the superlattices, with specific energies listed in Supplementary Table 3. For example, in SL27, the 37-meV peak is at a lower energy in the STO compared with the CTO, but the 60-meV and 97-meV peaks are at higher energies, with the interfaces exhibiting intermediate values, demonstrating the capacity to measure changes induced by local atomic displacements. Furthermore, we note that, although the interface and total-structure spectra bear some similarities, there are features in the interface spectra that cannot be reproduced by a mixture of the bulk-like neighbouring phases (Supplementary Fig. 11a, b). Unique vibrations emerging from the octahedral coupling at the interface must be present to account for the discrepancy between the total and interface spectra, as calculated via DFT. Thus, we demonstrate that variations in the localized vibrational spectra are ascribed to the regions of differing symmetry, namely, the STO, CTO and structurally diffuse interfaces.

Like the octahedral-tilt variation, spatial variations in the EELS response reduce with decreasing period. The spectral similarity from layer to layer in SL4 relative to SL27 and the exact match between the total and interface spectra indicate that the vibrational state of the

superlattice is approaching that of the interfaces, demonstrating the importance of local vibrational structure as length scales decrease. The global response of the interface vibrations further demonstrates the importance of local vibrational structure as length scales decrease.

The predominance of these interface vibrations and their global response has been shown to affect the thermal characteristics of STO–CTO superlattices, where a crossover from incoherent to coherent phonon transport is observed as the heterostructure periodicity decreases[10]. Previous reports have suggested that reduced zone folding of phonon dispersion leads to an increased group velocity, but direct evidence connecting underlying phononic processes, new modes and structure to the macroscopic transport mechanisms remains lacking[10,37]. To connect the localized interface modes and structural transitions observed above to a macroscopic response, we perform Fourier transform infrared spectroscopy (FTIR) (Fig. 4a)[10,15–18]. The residuals from a linear combination of STO and CTO films are quantified to accentuate changes unique to the superlattice. Owing to selection rules, the spectra of STO contains fewer reflectivity minima relative to CTO. As the STO and interface incorporate CTO tilt patterns, further loss of reflectivity should occur[38]. Residuals are observed near 500 cm$^{-1}$ and 560 cm$^{-1}$ that decrease with decreasing period thickness (Fig. 4a). FTIR spectra from a sample on $(LaAlO_3)_{0.3}(Sr_2TaAlO_6)_{0.35}$ substrates and ultraviolet-Raman experiments show similar trends, as shown in Supplementary Figs. 14–16. DFT informs that infrared-active phonons emerge in the superlattices and are associated with layer-localized, layer-delocalized and interface-localized Ti–O vibrational modes that are not infrared-active in bulk STO or CTO (Supplementary Fig. 8). The residual responses are similar in energy to those observed locally with both EELS and DFT. In addition, the sum of residuals (Supplementary

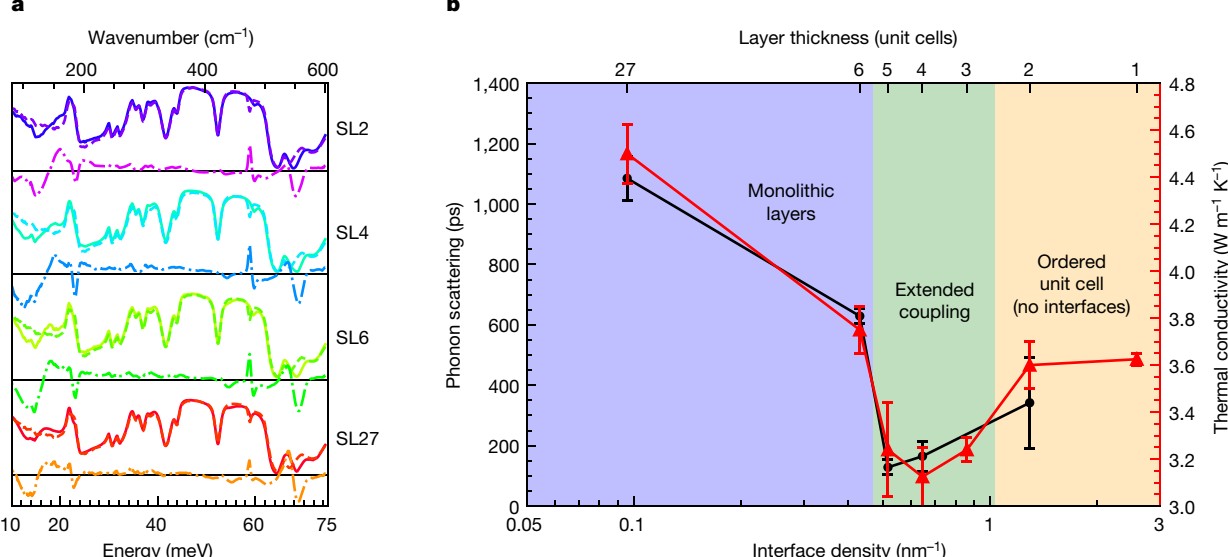

**Fig. 4 | FTIR and TDBS response of the STO–CTO superlattices. a**, Raw (solid), fitted (dashed) and residual (dot-dashed) data for FTIR from the superlattices on an NGO substrate. The 200-nm STO and CTO thin films on NGO substrates used to fit the superlattice spectra are shown in Supplementary Figs. 13b, 15e, f. The difference curves are scaled by a factor of two for clarity. **b**, Phonon lifetimes (black squares) as measured via TDBS compared with the thermal conductivity (red triangles, from ref. [10]) of STO–CTO superlattices with varying periodicities. The strong correlation between the two techniques conclusively demonstrates a transition in phonon scattering rates across the structural transitions elucidated with STEM/EELS. The error bars represent the standard deviation.

Fig. 17) scales with interface density. We conclude the residuals are a consequence of the local displacements existing at the interfaces of the superlattice.

To directly investigate the emergent phonon dynamics, we perform time-domain Brillouin scattering (TDBS) measurements, which detect the propagation of the zone-centre longitudinal modes as a function of time, thus providing a measure of their lifetime[39,40]. For short-period superlattices (SL2), an increase in both the phonon scattering time and the thermal conductivity is observed; this is no longer described by a superlattice with interfaces, but rather has a single structure with uniform octahedral tilts. We find the previously reported thermal conductivity to be correlated with the phonon scattering times (Fig. 4), with both having a minimum at SL4[10]. Thus, in strong agreement with our STEM, EELS and SHG results, the phonon lifetime is found to increase as material interfaces vanish. This combination of structural, electronic/optical and vibrational characterization techniques unambiguously demonstrates the underlying coupling of heterostructures and emergent global properties that are driven by interfaces.

## Conclusions

From a broader perspective, these results provide an alternative pathway by which nanostructuring can influence material properties. Typically, a superlattice response is thought to arise through either localized or coherent effects. The latter concerns the coherence length of the states with respect to interface periodicity, whereas for localization, discrete confined quantum states exist that are different to those in the bulk. Neither of these views explain the changes in the vibrational response observed here, because they neglect underlying symmetry changes that can propagate into the constituent materials. When scaling the phases that constitute the material to unit-cell dimensions, the solid takes on a new symmetry that cannot be explained by a combination of the constituent materials. In these STO–CTO superlattices, this new structure results from octahedral coupling between the layers. Here we have directly imaged these localized changes in symmetry and their impact on vibrations using a combination of STEM iDPC and

monochromated EELS, with the conclusions drawn supported by DFT. We have further demonstrated how the observed localized phenomena evolve from locally affecting the superlattice at larger periods to dictating the global response of the superlattice as the period decreases via nonlinear optical and phonon lifetime measurements. It is important to note that the reported changes in symmetry are not from the global periodicity of the superlattice. Rather, it is the local symmetry changes at the interfaces, and their spatial distribution, that ultimately dictate the entire macroscopic response of the solid as the period thickness decreases. Therefore, tailoring interfaces, and knowing their local response, provides a means of pursuing 'designer' solids with emergent infrared and thermal responses not inherent within either of the constituent bulk materials.

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

## Methods

### Thin-film growth

Superlattices grown on $NdGaO_3$ (NGO) substrates were synthesized using reflection high-energy electron diffraction assisted pulsed laser deposition. Further discussion of the growth is found in ref. [10] and its supplementary information. Superlattices grown on $(LaAlO_3)_{0.3}$ $(Sr_2TaAlO_6)_{0.35}$ (LSAT) substrates were realized via hybrid molecular beam epitaxy as outlined in ref. [41].

### Electron microscopy

Selected-area diffraction patterns are from a Thermo Fisher 80–300 kV Titan operating at 300 kV and equipped with a Gatan OneView camera.

Scanning convergent-beam electron diffraction, ADF and iDPC images were acquired on a Thermo Fisher Themis Z-STEM operating at 300 kV. ADF and iDPC images were acquired with a convergence angle of 30 mrad, a probe current of 200 pA and a dwell time of 625 nm $px^{-1}$. A 145-mm camera length projected onto the ADF detector with a 200-mrad outer radius and 40-mrad inner radius. The segmented ADF detector used for iDPC had a 38-mrad outer radius and a 10-mrad inner radius. The iDPC images (Fig. 1f, i, l) allow the measurement of the positions of the O and Ti columns, enabling quantification of the octahedral tilt angles[42,43]. The position of the metal sites was refined by thresholding, finding the centre of mass, then fitting with two-dimensional Gaussians. The spacing of the O columns necessitated locating the atomic columns manually.

Vibrational EELS spectra were acquired at an operating voltage of 60 kV using a Nion HERMES monochromated aberration-corrected dedicated STEM with a convergence angle of 32 mrad, an entrance aperture collection angle of 25 mrad and an energy dispersion of 0.413 meV per channel for the SL2 and SL4 acquisitions and 0.826 meV per channel for the SL27 acquisition, and the achieved energy resolution varied between 12 meV and 17 meV (Supplementary Fig. 12). In this study, all EELS spectra were acquired in an off-axis mode, obtained by shifting the electron diffraction pattern with respect to EELS entrance aperture. In EELS, delocalized dipole scattering can dominate signals and mask local variations in phonon populations, which can be detected by impact scattering. However, by acquiring EELS from only electrons scattered out to high angles, the dipole scattering is reduced more than the impact scattering and the localized signals can be retrieved[29,31]. Here we displace the optic axis from the EELS entrance aperture by about 50 mrad in the non-dispersive axis of the spectrometer, and only integrate pixels in the top half of the acquired signal. Thus, the off-axis EELS shown in this study has an effective collection semi-angle of about 12.5 mrad that is scattered about 55.5 mrad from the central optic axis. By integrating the signal at higher angles, we preferentially select vibrations that have undergone impact scattering, which is a more spatially localized signal, and exclude the electrons that have undergone dipole scattering to low angles, which is spatially delocalized.

Vibrational EELS background removal with fitted functions can introduce error because the realistic background does not have a functional shape, owing to overlapping of the zero-loss peak and real spectral features such as non-resolvable acoustic or low-energy optic phonons. Therefore, we choose to take an alternative approach, normalizing the intensity of the spectra by multiplying by the energy squared ($E^2$), thus uniformly normalizing the spectra to a quadratic background. To increase the signal-to-noise ratio and for comparison with theoretical predictions, we take the average of all spectra in STO, CTO and the interface layers, akin to layer DOS in DFT. The interface signal is defined as one unit cell in length for consistency with structural characterization. Interfaces are assigned using the second derivative of the off-axis high-angle annular dark-field signal. An example of interface assignment for SL4 is shown in Supplementary Fig. 9. The layer-averaged signals from each layer can then be easily compared with one another and the average superlattice signal. An initial concern of qualitatively comparing the vibrational response of the layers and interfaces was possible thickness-dependent trends in scattering probability relative to other excitations. To eliminate these effects, we compare the thickness dependent trends (Supplementary Fig. 10) and find that the layer-averaged signals converge after a few layers. We use the layers after signal convergence for comparing superlattices. In SL2, structural characterization showed that we cannot define a structural interface or structurally unique layers. We therefore had three choices: (1) define the entire period as a layer, which is the same as the average superlattice signal and provides no comparable spectra; (2) revert to the definition for chemically defined interfaces, which provides STO and CTO layer-average spectra for comparison with each other and the average superlattice signal; or (3) use the structurally diffuse interface width of one unit cell, which provides the interface, STO and CTO layers. The third definition does not leave an actual STO or CTO layer, because only a single atomic plane of $TiO_2$ remains between the interfaces. The lack of a complete STO and CTO layer was part of the rational for a single phase in the iDPC analysis, making the third choice inconsistent with the structural analysis. The second choice would be inconsistent with the EELS analysis of SL27 and SL2. We chose the third choice so that the EELS analysis between the three superlattices was consistent, and because atomic-resolution conditions were not used in all EELS experiments, making the one-atomic-plane delineation of layers infeasible. The lack of spectral change from the STO layer to the interface layer to the CTO layer observed in the EELS analysis of SL2 then shows that the layers behave similarly, which is consistent with each having a similar symmetry.

### Calculations

The DFT calculations used the Vienna Ab initio Simulation Package (VASP)[44] with the projected-augmented wave (PAW)[45,46] method and the local-density approximation (LDA)[47]. Phonon calculations were performed using the LDA for exchange correlation because it has been found to perform better for phonons at the Γ point, which is of interest here, in bulk CTO and STO[48–50]. The plane-wave basis energy cut-off is 600 eV. The superlattice structural models were constructed by alternatively combining *Pbnm*-phase STO and CTO in the *c* direction with specific thicknesses. An SL8 model was chosen to obtain tilt angles for a large-period superlattice, knowing that the interface coupling is limited to a few atomic planes and the prohibitive computational requirements for simulating SL27. For structural relaxation, the structures were relaxed until the atomic forces were less than 0.01 eV $Å^{-1}$. The lattice parameters were also optimized for each superlattice model. Phonon calculations were performed using the finite-difference method. For structural relaxation and phonon calculations, the *k*-samplings are 6 × 6 × 6 for bulk STO and CTO, 4 × 4 × 2 for SL2 and SL4, and 2 × 2 × 2 for SL8. A full-width at half-maximum of 16 $cm^{-1}$ was used to plot the projected PDOS.

The PDOS of each model was obtained by performing a weighted average over the respective constituent layers. The projected PDOS of the respective constituent layers was normalized by the number of atoms per layer to provide a consistent comparison between the three superlattices. The total PDOS of each model was then obtained by $n_{total} = (x \times n_{STO} + y \times n_{CTO} + z \times n_{int})/(x + y + z)$, where $x$, $y$ and $z$ are the numbers of atoms considered in each layer. In particular, the PDOS of SL27 was obtained by averaging the phonon modes of the intrinsic bulk STO, the intrinsic bulk CTO and the SL8 interface. The interface in all models is defined as one unit cell on either side of the chemically defined interface, which is consistent with experimental and calculated structures. As the primary structural changes are associated with $TiO_6$ octahedra, we can assume that the distinct vibrational state of different superlattices are primarily contributed by Ti/O-related vibrational modes. Therefore, we project the PDOS on Ti and O atoms, emphasizing the symmetry–phonon relation. As only the phonon modes parallel to fields are activated, the PDOS is also projected in the (110) plane that is perpendicular to the electron beam. For completeness, we have also projected the PDOS on the A-site atoms, which can be found in Supplementary Fig. 7.

## Optical spectroscopies

Raman spectroscopy was performed on samples synthesized atop NGO with a Horiba LabRam Raman instrument employing a 325-nm laser focused using a ×40/0.5 numerical aperture objective. Laser powers were verified to be inconsequential to the results. At this wavelength, the skin depth for the exciting ultraviolet light is 26 nm within STO whereas it is more than 1 μm for CTO. Despite the transparency of CTO, all Raman-examined films were 200 nm in thickness and thus contain at least 100 nm of STO. This is more than three times the skin depth and thus the underlying NGO does not affect the Raman experiment. The monolithic samples show a response expected from their bulk form[48,51–57]. Raman and FTIR spectra were fitted using a least-squares minimized linear combination of acquired monolithic spectra. This extenuated the differences between the superlattice and the constituent materials and helped remove the substrate response in the FTIR.

## Second-harmonic generation

SHG measurements were performed on SL27, SL6, SL4, SL3, SL2 and SL1 with nominal thicknesses of 200 nm atop NGO substrates. In contrast to linear optical measurements, which are dictated by the average of the linear response of the materials comprising the superlattice[58], the higher-rank dielectric tensor associated with SHG vanishes if the constituent materials have inversion symmetry. The home-built SHG microscope is centred on a 1,040-nm neodymium-doped yttrium orthovanadate, roughly 100-fs Gaussian laser source that is focused to the sample surface at an incident angle of 45° relative to the surface normal using a ×10 microscope objective (numerical aperture 0.28). The incident beam polarization is rotated using a half-wave plate. The forward-scattered beam, containing both the fundamental and second-harmonic frequencies, is collected with a lens. Through a series of band-pass filters, non-second-harmonic components are filtered out, whereas the second-harmonic component is focused to an amplified avalanche photodiode. The generated voltage is further amplified via lock-in detection demodulated at the laser repetition rate. The reported second-harmonic values are the parabolic coefficient determined by fitting the measured SHG intensity (for example, lock-in photodiode response) as a function of incident laser power for each sample. The square dependence of measured intensity versus incident field indicates that no higher harmonics are measured or that no optical leakage of the fundamental frequency is reaching the detector.

## Time-domain Brillouin scattering

The TDBS measurements were performed using an 80-MHz, 800-nm titanium:sapphire oscillator (about 100-fs pulses) that is split into two optical paths before reaching the sample. The first beam is used as a high-energy pump pulse, that, when focused to the sample surface, stimulates coherent acoustic phonon modes via rapid thermal expansion of the material. This pump pulse is frequency doubled (400 nm) for these measurements to increase optical absorption in the STO–CTO layers. The second beam is sent down a mechanical delay stage to vary the time at which the pulse reaches the sample surface; this low-energy 'probe' pulse monitors changes in the optical properties of the sample following excitation as a function of time delay between the two pulses. As the coherent longitudinal phonon mode propagates through the superlattice, the probe beam partially reflects off the sample surface and partially off the coherent wave. The distance between these partial reflections evolves in time owing to propagation of the phonon mode, and thus operates as a Fabry–Perot interferometer, where for distances that are integer multiples of the probe wavelength, constructive interference is observed in the signal, and for half-integer wavelength distances, the two reflections destructively interfere and reduce the signal. The temporal decay of these sinusoidally varying oscillations is a direct monitor of the lifetime of the pump-generated longitudinal vibrational mode within the superlattic structure.

## Data availability

The datasets generated during and/or analysed during the current study are available from the corresponding authors on reasonable request. Please contact E.R.H. regarding STEM imagining data. Please contact E.R.H. or J.A.H. regarding EELS data. Please contact S.T.P. regarding DFT data. Please contact P.E.H. regarding SHG and Brillouin zone scattering data. Please contact E.R.H. regarding Raman or FTIR data.

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

**Acknowledgements** E.R.H., J.A.T., S.M. and P.E.H. appreciate support from the Office of Naval Research through a MURI Program, grant number N00014-18-1-2429. J.A.T., S.M. and P.E.H. acknowledge support from Army Research Office, grant number W911NF-21-1-0119. Use of the Thermo Fisher Scientific Z-STEM and Titan instruments within UVa's Nanoscale Materials Characterization Facility (NMCF) was fundamental to this work. Theory at Vanderbilt University was supported by the US Department of Energy, Office of Science, Basic Energy Sciences, Materials Science and Engineering Directorate grant number DE-FG02-09ER46554 and by the McMinn Endowment. Calculations were performed at the National Energy Research Scientific Computing Center (NERSC), a US Department of Energy Office of Science User Facility located at Lawrence Berkeley National Laboratory, operated under contract number DE-AC02-05CH11231. The oxide heteroepitaxy synthesis work at Berkeley and Penn State is supported by the Quantum Materials program from the DOE Office of Science, Basic Energy Sciences, under contract number DE-AC02-05CH11231, and by the MRSEC for Nanoscale Science at Penn State through grant DMR1420620. EELS experiments conducted as part of a user proposal at the Center for Nanophase Materials Sciences, which is a DOE Office of Science User Facility using instrumentation within ORNL's Materials Characterization Core provided by UT-Battelle, LLC, under contract number DE-AC05-00OR22725 with the DOE and sponsored by the Laboratory Directed Research and Development Program of Oak Ridge National Laboratory, managed by UT-Battelle, LLC, for the US Department of Energy. R.R. also acknowledges the ARO MURI under agreement W911NF-21-2-0162. J.R. acknowledges support from the Army Research Office with award numbers W911NF-19-1-0137 and W911NF-21-1-0327. J.D.C. and J.R.M. acknowledge funding under NSF, Division of Materials Research award number 1904793. S.M. acknowledges support from the NIH Biotechnology Training Program.

**Author contributions** E.R.H., J.A.H. and J.M.H. contributed to the acquisition, analysis and understanding of all scanning electron microscopy data. D.-L.B., A.O. and S.T.P. contributed all density functional theory calculations and corresponding analysis. S.M. and J.A.T. contributed acquisition, analysis and understanding of the SHG and TDBS spectroscopy data. Z.T.P. and T.E.B. contributed acquisition analysis, and understanding of ultraviolet-Raman data. J.R.M., T.E.B. and J.D.C. contributed acquisition, analysis and understanding of Fourier-transform infrared spectroscopy data. A.K.Y., R.C.H., R.E.H., J.R. and R.R. contributed growth expertise and samples used in the analysis. J.F.I. and P.E.H. contributed an understanding of how the crystal and vibrational structure impacted broader material properties.

**Competing interests** The authors declare no competing interests.

**Additional information**

**Correspondence and requests for materials** should be addressed to Eric R. Hoglund, Jordan A. Hachtel, Sokrates T. Pantelides, Patrick E. Hopkins or James M. Howe.
