## [Peer Review File · Nature]

Manuscript Title: Emergent Interface Vibrational Structure of Oxide Superlattices

Reviewer Comments & Author Rebuttals

Reviewer Reports on the Initial Version:

Referee #1 (Remarks to the Author):

Report Nature manuscript 2021-05-08011

Dr Hoglund and co-authors entitled, "Nanoscale Phonon Spectroscopy Reveals Emergent Interface Vibrational Structure of Superlattices"

The manuscript by Dr Hoglund and co-authors entitled "Nanoscale Phonon Spectroscopy Reveals Emergent Interface Vibrational Structure of Superlattices" addresses the important question of how interfaces are constructed in complex perovskite-type heterostructures, how they impact macroscopic physical properties and how different types of heterostructures can allow the tuning of such macroscopic properties. The article provides strong evidence for changes of the structural symmetry and phonons at the interface.

Most experimental studies in literature address the question of the atomic arrangement by x-ray and electron diffraction techniques and the microstructure by different modes of AFM. The present study has the originality to address such questions by the experimental and theoretical analysis of phonons, which can determine physical properties, and which provide a dynamic view on atomic arrangements through the coherence of phonons. This originality has the potential to influence the field and the view on superlattices (or heterostructures), where interfaces play an increasingly important role.

The paper is original and provides significant new results through an original approach. However, I consider that it addresses a more specialized readership than Nature, I thus suggest considering another journal of the Nature publishing group.

The following multiple points should be pondered before considering publication.

Even though the approach as such is novel, the study extends earlier observations of phonons in the ultrathin single film regime, which I suggest considering for the interpretation of data and for reference:

- D. A. Tenne et al., "Probing nanoscale ferroelectricity by ultraviolet Raman spectroscopy," *Science* 313, 1614 (2006).
- L. Vattuone et al., "Phonons in thin oxide films," in *Oxide Materials at the Two-Dimensional Limit* (Springer, 2016), pp. 169–199.
- J. Prempere et al., "Surface stress and lattice dynamics in oxide ultrathin films," *Phys. Status Solidi B* 2020, 257, 1900650
- A. Schober et al. "Vibrational properties of LaNiO₃ films in the ultrathin regime" *APL Mater.* 2020, 8, 061102

Also, the article would gain from being set it into a larger context, by considering that interfaces in superlattices are conceptually similar to phase boundaries, homo- and heterointerfaces, domain walls or other quasi-2D objects for which exceptional properties and coupling phenomena have been reported (G. F. Nataf et al. *Nature Reviews Physics*, 2, 634–648 (2020)). The concept of "transitional regions" appears as relevant (Everhardt, A. S. et al. Temperature- independent giant dielectric response in transitional BaTiO₃ thin films. *Appl. Phys. Rev.* 7, 011402 (2020)).

The manuscript claims that the presented interface tuning has the potential to tailor physical

properties, with a particular mentioning of infrared and thermal responses. Even though this is highly likely, this claim should be supported by measurements of the physical properties of the investigated films. Here again, the narrative would be stronger by indicating that electronic and magnetic properties of perovskite-type superlattices are strongly affected by the octahedra tilt angle.

Specific comments :

- Page 2: The first sentence this should be toned down as this is known for strained single thin films
- Page 2: Wording "Symmetries atypical of either constituent" is not clear
- Page 2: I suggest toning down the sentences between materials and interfaces, which are known to the community.
- Page 2: precise what you mean by "spatial resolution". Lateral and or depth?
- Page 3 and following parts: I suggest describing the potential limitations of DFT calculations for the phonon response of the investigated nano-features. Has the interface a defined symmetry?
- Page 3: What is meant by "length scale of octahedral coupling", please explain.
- Page 4 and following: Please be precise what unit cell means (the pseudo-cubic unit cell or the tilt-induced cell-doubling unit cell)
- Page 4 & 5: Would it be possible to estimate the octahedra tilt angle from the intensity of the superstructure reflections?
- Page 4 & 5: Can you please say something about potential diffuse scattering which could indicate the loss of translation symmetry or changes in the coherence. Is the width of the superstructure reflection the same for different families?
- Page 5: "the orientations of octahedra tilts are different". What's about the amplitude of the tilts?
- Page 5 "single crystal pattern". What does this mean? What about the FWHM of the reflections?
- Page 6 "Fm-3m" : do you mean the simple cubic structure "Pm-3m" ? Or do you infer chemical ordering?
- Page 7: The term "crystal of interfaces" is unusual and likely not correct. Same for "crystalline hybrid" on page 10. Two unusual terms for the same object. Since the authors address phonons, it could be useful to reflect on the two proto-type behaviors of mixed systems: virtual ion behavior and two-mode behavior. What is the observed regime?
- Page 7 and others: A reminder of how crystal structures vary in the STO-CTO solid solution could be insightful.
- Page 8: The interpretation of the DOS is at the hard limit of overinterpretation. For instance a peak shift does not formally indicate a change in symmetry, but could also be a shift of atoms within a given space group, as known for so-called soft modes.
- Page 9, Figure 2(k). I guess we should read here SL2 and not SL4
- Page 11: The question "Do the interface vibrations .." is trivial and the answer "Yes" is known and accepted in the community.
- Page 11, Figure 3. Title: "Raman" should be added as not only IR data are shown. I suggest adding a reference spectrum of NGO to support the statement that the substrate is not seen by Raman scattering, same for IR. Title of (c) should read "FTIR LSAT". An example of the raw, fitted, residual data should be given in more detail and with a zoom in the supplementary material to allow a proper and critical inspection, which is not possible with the current Figure 3. Only with this a referee or author can further judge on the described analysis and interpretation.
- Page 11: The "emergent Raman response" near 220 and 500 cm^{-1} should be interpreted in terms of the expected vibrational pattern expected in the respective spectral region (tilt, cation displacement, breathing etc.). Between 200 and 500 cm^{-1} the overall second order Raman signature is much stronger, this might mask other features.
- Page 16/17: The UV light is thus very differently absorbed by STO and CTO, thus the response for a given small volume must be very (!) different is the described decomposition of the spectra under this condition still stable. Please give the thickness of the investigated monolithic films.

Referee #2 (Remarks to the Author):

In the manuscript "Nanoscale Phonon Spectroscopy Reveals Emergent Interface Vibrational Structure of Superlattices", Hoglund et al. use imaging techniques and electron energy loss spectroscopy in the (scanning) transmission electron microscope to show evidence for an emergent vibrational response in oxide superlattices. The observation that, with decreasing superlattice layer thickness, the local vibrational response converges towards that of the layer interfaces, is attributed to changes in atomic scale symmetry uncharacteristic of either "bulk" constituent material. Local microscopy measurements are supported by theoretical modelling and global measurements using Raman and FTIR optical spectroscopies. The results are overall convincing and support the authors conclusions. Key to this work is the combination imaging and spectroscopy techniques, providing novel insight into emergent vibrational properties of solids down to the atomic scale. These results are likely of great interest to a wide audience.

However, before I can recommend this manuscript for publication, the following points should be addressed by the authors.

While the data presentation is good overall, the plot legends for the black-line plots in figures 2(k) and S10(f) be "SL2" not "SL4". Moreover, should the plot title in fig. 3(c) should be "FTIR LSAT" and not "LSAT NGO"?

Figure 1: the authors use red, green, and blue "atoms" to label atomic column positions without clearly defining what these represent. Adding labels to the figure and/ or a description of this in the figure caption would be appropriate.

On lines 225-227, the authors state: "...there are features in the interface spectra that cannot be reproduced by a mixture of the bulk-like CTO and STO phases (shown in more detail in supplementary Figure S10(a,b))." As emergent properties are the focus of this manuscript, the authors should elaborate on this point.

Figure 3: while the residual Raman spectra clearly show an increasing intensity with decreasing layer thickness, this is not so obvious for the IR measurements. It would be interesting to attempt to quantify this difference, helping the reader better grasp the magnitude of this effect. This would also help substantiate their claim that FTIR residual spectra show a "similar evolution" to that of the Raman residual spectra. While there may be many ways of going about this, one suggestion is to plot the summed absolute values over regions of interest and/ or the full range in the residual spectra shown in figure 3, as a function of layer thickness, for both Raman and IR.

In section 5.2, on line 311, the authors state that they "only integrate pixels in the top half of the acquired signal". I assume here they are referring to the "top half" of the spectrometer camera, as displayed on the computer screen. It is not obvious from the text why the authors made this choice, thus a justification should be given.

Author Rebuttals to Initial Comments:

1 Reviewer #1 (Remarks to the Author):

Report Nature manuscript 2021-05-08011

Dr Hoglund and co-authors entitled, "Nanoscale Phonon Spectroscopy Reveals Emergent Interface Vibrational Structure of Superlattices"

The manuscript by Dr Hoglund and co-authors entitled "Nanoscale Phonon Spectroscopy Reveals Emergent Interface Vibrational Structure of Superlattices" addresses the important question of how

interfaces are constructed in complex perovskite-type heterostructures, how they impact macroscopic physical properties and how different types of heterostructures can allow the tuning of such macroscopic properties. The article provides strong evidence for changes of the structural symmetry and phonons at the interface.

Most experimental studies in literature address the question of the atomic arrangement by x-ray and electron diffraction techniques and the microstructure by different modes of AFM. The present study has the originality to address such questions by the experimental and theoretical analysis of phonons, which can determine physical properties, and which provide a dynamic view on atomic arrangements through the coherence of phonons. This originality has the potential to influence the field and the view on superlattices (or heterostructures), where interfaces play an increasingly important role.

The paper is original and provides significant new results through an original approach. However, I consider that it addresses a more specialized readership than Nature, I thus suggest considering another journal of the Nature publishing group.

To begin, the authors would like to sincerely thank Reviewer 1 for their very in-depth review of our manuscript. The additional references to prior works provided by the reviewer are greatly appreciated and have been included and contextualized in our revised manuscript. Additionally, the various comments and suggestions address a number of essential aspects in our work that were perhaps lacking in our initial submission; we believe that the revisions described below have strengthened our manuscript and allowed us to connect our findings to the scientific community at large with significantly increased clarity. These revisions increase both the impact and breadth of our manuscript.

The following multiple points should be pondered before considering publication.

Even though the approach as such is novel, the study extends earlier observations of phonons in the ultrathin single film regime, which I suggest considering for the interpretation of data and for reference:

We appreciate the reviewer's comment regarding the novelty of our approach, as well as the provided references that contextualize current studies of phonon/vibrational properties of materials in the ultrathin single film regime. We believe the novelty of our approach is partly what differentiates this study from these prior works and warrants dissemination to the scientific community at-large. Specifically, in contrast to previous studies, we report on direct measurements elucidating upon interface-driven vibrational phenomena that has, to date, required indirect measurements, i.e. measurements with photons, neutrons, etc. and carefully controlled film-thickness. In other words, our work provides the highest spatial-resolution measurement of interfacial vibrational properties of materials to-date. Prior measurements solely rely on controlling the size of materials to decouple interface vibrations from bulk vibrations. For example, by controlling film thickness the relative contribution of interface and bulk phonons can be decoupled. However, as the reviewer aptly notes in a later comment, the structure of materials often evolves as film thickness changes; for example, strain relaxation of superlattices above a critical thickness leads to significant changes in their properties below such a thickness threshold. These extrinsic material modifications will ultimately

change the nature of interface and bulk vibrations. The two-sided relation that could impact interpretation does not occur in our work, because there is no need to scale any part of the system to determine vibrations or structures. Our probe *directly* interrogates the interface and can measure its local structure and vibrations without convolution of extrinsic factors. We have included the suggested references in multiple locations as they enhance the impact of this work.

The provided references (with the exception of D. A. Tenne *et al.*) detail the effect of interface phonons on macroscopic response by measuring phonons or structure as a function of thin-film thickness. From the change in response versus thickness they indirectly interpret the effect of interfaces, and using theory, known symmetries, and selection rules can assign specific interface phonon modes. This is a key point explicitly stated in the conclusions of L. Vattuone *et al.* Here, we do not have the necessity for thickness variation, known selection rules, or theory. We directly observe the presence of unique interface vibrations by ‘placing’ our electron probe at the interface and monitor the interfacial response. At present, we lack the spectroscopic resolution provided by other techniques, but the spatial resolution provided is the novelty of the technique. This is, in part, the power of vibrational EELS in a STEM and one of the differentiating components of our paper. A vital component that separates our paper from prior studies is that our study is performed on superlattices, or periodic heterostructures. Total film thickness variation cannot be used in a superlattice due to their periodic nature. Variation of the layer thickness also cannot be used, because as we show, structural changes may occur within the layers that will directly impact the total vibrational response.

Such structural changes were also shown in the provided reference D.A. Tenne *et al.* for polarization rather than octahedral tilt. Their work however is also distinctively different because it entirely concerns the structure of the SrTiO₃ and does not make a connection with interface phonons. The authors included a couple sentences discussing this as we believe our experiments lend light on their work. Thank you for pointing us to the reference. As emphasized in our manuscript, the novelty of this work is that we have directly and locally observed interface vibration using vibrational EELS, which we directly connect to the structure by imaging the atomic columns in layers and interfaces. We then go further and explain the macroscopic trends observed in our Raman and similar works with our local observations.

Further explanation for each reference to follow.

- D. A. Tenne et al., “Probing nanoscale ferroelectricity by ultraviolet Raman spectroscopy,” *Science* 313, 1614 (2006).

The reference D. A. Tenne et al., “Probing nanoscale ferroelectricity by ultraviolet Raman spectroscopy,” *Science* 313, 1614 (2006) observe the polarization of thin SrTiO₃ layers in a BaTiO₃-SrTiO₃ superlattices by measuring the global temperature-dependent UV-Raman response of a polar phonon mode. The authors agree that this is a great reference to include because it describes the incorporation of the polar atomic displacements of BaTiO₃ in SrTiO₃ as layer thickness decreases, which is very similar to our observation of SrTiO₃ inheriting tilts from CaTiO₃. The paper also makes a connection between phonon modes that cause the observed atomic displacements. Given the reliability, the authors have included D. A. Tenne et al. to the list of reference and included a brief discussion on page 2, 7, and 14.

We would like to also point out how our research differs from the provided reference. The work by D. A. Tenne et al. measures 2x13 and 8x4 unit-cell BaTiO₃-SrTiO₃ superlattices using UV-Raman, which measures the collective response of the thin-films. The work informs that small layers of SrTiO₃ are polarized, but it does not demonstrate the origin of polarization as it lacks sensitivity to spatial changes of the polar phonon modes. As shown in our current work, the phonon modes, and related displacements, can change as a function of position in the SrTiO₃ layers. In our case we observe that the changes occurring in SrTiO₃ stem from the CaTiO₃-SrTiO₃ interfaces, and the tilts are not uniform. The structural displacements and TO₂ phonon modes of SrTiO₃ in the BaTiO₃-SrTiO₃ superlattices could be uniform throughout the SrTiO₃, or they could be non-uniform and associated with coupling at interfaces. In our article, the capability to spatially resolve the vibrations and structure with relation to the interfaces in the material is key. Such spatial resolving capabilities allows the authors to not only show that the vibrations and structure of superlattices change when layer thickness decreases, but also allows the authors to show that the interfaces and the length-scale of interfacial changes are what cause transitions.

The authors have added the following three suggested references and pointed to the novelty of our work and how it informs prior measurements.

- L. Vattuone et al., “Phonons in thin oxide films,” in *Oxide Materials at the Two-Dimensional Limit* (Springer, 2016), pp. 169–199.

The review by L. Vattuone et al. is another great reference. Thank you for bringing the authors attention to this review. It is relatable to our current work and will be helpful for future endeavors. The review discusses changes in the vibrational response of thin-film oxides using techniques like high-resolution electron energy-loss spectroscopy and infra-red absorption spectroscopy with relation to both chemical and structural changes. The majority of thin-films compared have different stoichiometry and polymorphs or are related to surface reconstruction. In general, a key takeaway from the reference is that ultra thin-films are different than bulk materials. Our paper is similar because we show that the large-period superlattice (bulk-like) is not like the short-period superlattices (ultra thin-like), but the cause for the differences are starkly different because we do not have changes in stoichiometry and the structural differences occurring are minor compared to the polymorphic transitions and reconstructions reviewed. The authors also point to the conclusion of this reference:

“The assignment of the observed vibrational signatures is non trivial and is ultimately based on symmetry arguments and on the selection rules operative for the probe particles (electrons or photons), for the particular substrate (metallic or not) and for the given scattering geometry. While for thick enough thin films the predictions of macroscopic dielectric theory are generally confirmed by experiments, in the ultrathin limit microscopic modes are also observed and for their assignment a theory is usually required.”

This again points to a strength of our work. Although modeling strengthens our observations and helps us understand the structural-vibrational relations more in depth, modeling and selection rules are not entirely necessary. The current work does not require symmetry and selection rules. We can directly observe the structural and vibrational changes and they are of course related.

- J. Premper et al., "Surface stress and lattice dynamics in oxide ultrathin films," Phys. Status Solidi B 2020, 257, 1900650

The suggested reference by J. Premper et al. discusses the vibrational response of NiO on Ag, BaO on Ag, and BaTiO₃ on Pt as a function of thin-film thickness using high-resolution electron energy-loss spectroscopy and low-to-medium energy electron diffraction. Thank you again for the reference. The structure-vibration relation in the limit of thin-films is very applicable to our current work. Strain is, in effect, a structural displacement and should affect phonons. This is clearly shown in Figure 10 of the suggested reference where J. Premper et al. show a linear relation between the Fuchs–Kliewer phonon energy and film strain. This is much like our work showing the relation between octahedral tilting and the observed changes in vibrational energy.

Our work although very much related is also different from their work. The strain in the ultra thin-films is expected to be uniform and J. Premper et al. show that increasing film thickness relaxes strain. This is consistent with decades of film growth and characterization. J. Premper et al. then connect the stressed/strained state of the film with various vibrational modes in the thin-film. These findings are very insightful and worth referring to in our work. The relation between the uniformly distributed stress or strain to vibrations inherently implicates that the vibrations must also be uniformly affected. We show that this is not necessarily the case. Local atomic displacements and symmetries influence local vibrations. Important details that determine the state of the material system are lost if interfaces are neglected. The change in vibrational response of the film that they observe during growth can be thought of as an average over the local interface and intrinsic strain modified film vibrations.

This is an excellent reference for potential readers and as such has been included.

- A. Schober et al. "Vibrational properties of LaNiO₃ films in the ultrathin regime" APL Mater. 2020, 8, 061102

Again, this is a very great article that is very relatable to our work. Their confocal Raman approach is very informative. A. Schober et al. show that when a thin-film decreases below a critical thickness the vibrations are a hybrid of the film and substrate. The hybridization is much like what we observe in our short-period superlattices. However, because we have a superlattice, and not a few unit-cell thick film, the hybridization is supported over a much thicker film. This is a great and relatable reference, and as such has been included.

The authors would say that our current work helps explain trends in their data and does not simply extend the prior work. By spatially mapping the vibrations we can conclude that their observations of hybridization are a direct impact of interface vibrations being present throughout the entire film. Their work does not include characterization of the interface vibrations but instead categorizes film and substrate vibrations. We show that the presence of interface vibrations are as important as bulk vibrations when layer (or film) thicknesses decrease. Furthermore, they interpret the rotations in the film using the phonon energies. We show through imaging that the octahedral tilts may not be uniform. One way to interpret their data, considering our findings, is that they measure the average vibrational state of the thin-films and use that to determine the average octahedral rotation present in the thin-films hybridize with the substrate as film thickness decreases. The influence of local interface octahedral coupling and interface vibrations are then a larger fraction of the average state as film thickness decreases, but the hybridization they discuss always exists.

Also, the article would gain from being set it into a larger context, by considering that interfaces in superlattices are conceptually similar to phase boundaries, homo- and heterointerfaces, domain walls or other quasi-2D objects for which exceptional properties and coupling phenomena have been reported (G. F. Nataf et al. Nature Reviews Physics, 2, 634–648 (2020)). The concept of “transitional regions” appears as relevant (Everhardt, A. S. et al. Temperature- independent giant dielectric response in transitional BaTiO₃ thin films. Appl. Phys. Rev. 7, 011402 (2020)).

We appreciate the reviewer’s suggestion, and are in agreement, that a more generalized context would increase clarity to potential readers of our work. To address this, we have included a statement extending the interface in superlattices to those in all interphase and intergranular boundaries. The revised text can be found on page 3 and now reads as:

“The vibrations and coupling regions present at interfaces in superlattices, in a broader context, occur at other interphase and intergranular boundaries and can result in remarkable properties.^{3,14–}

29”

The manuscript claims that the presented interface tuning has the potential to tailor physical properties, with a particular mentioning of infrared and thermal responses. Even though this is highly likely, this claim should be supported by measurements of the physical properties of the investigated films. Here again, the narrative would be stronger by indicating that electronic and magnetic properties of perovskite-type superlattices are strongly affected by the octahedra tilt angle.

We greatly appreciate the reviewer’s comments and suggestion. The authors agree that the correlation between physical properties and underlying crystal/phonon structure is a critical aspect of this manuscript. In our original manuscript, we aimed to guide the reader to previous reports on tuning the thermal and infrared responses via superlattice periodicity, which is ultimately driven by structural changes. However, in agreement with the reviewer’s comment, direct measurements on the superlattices investigated in our manuscript would be significantly more compelling and enhance the narrative for potential readers. To address properties aspect, we performed a series of additional experiments to investigate changes in both the thermal, optical, and electronic properties of these superlattices as they transition through varying degrees of structural coherence.

First, to support out structural characterization via STEM and determine the underlying electronic properties of these heterostructures, we performed second-harmonic-generation (SHG) experiments. In contrast to linear optical measurements, which are dictated by the average of the linear response of the materials comprising the superlattice (e.g., interface responses are negligible), the higher order optical response tensor associated with SHG vanishes in both STO and CTO bulk layers. In other words, due to a break in inversion symmetry, this technique directly measures the electronic polarizability of bonds comprising the *interfaces* between the layers. As described in our revised manuscript on page 9, these experiments elucidate upon the dynamic evolution of in the electronic structure of STO/CTO superlattices, which we find to be directly related to the structural evolution associated with the interfaces. Stated differently, we observe a direct correlation between the electronic polarizability of

the STO-CTO structure and the interfacial coupling determined via EELS measurements in addition to providing further experimental evidence of the structural transitions in moderate- to short-period STO-CTO superlattices.

While the measured nonlinear optical response is directly coupled to the underlying structure, we note separate consideration of changes in the infrared optical properties due to structural transitions in our investigated superlattices. To this end, we have performed linear dielectric measurements in the infrared, namely, FTIR measurements. For a material system undergoing changes in its vibrational spectra, one would expect the phonon-associated dipole to dictate this frequency range of photons. Due to the reduced thickness of the materials investigated here, a vast fraction of the infrared optical activity is associated with the underlying substrate. Thus, we compare the residual difference between a bare substrate's FTIR response to that of the SL + film heterostructure. As shown on page 14 of our revised manuscript, we observe a consistent change in the IR optical response at ~ 500 and 560 cm^{-1} ; this response is strongly related to the periodicity of the superlattice. Though the exact mode responsible for this feature is difficult to assign, it demonstrates the relationship between interfacial structural coupling and the optical response of our materials, in, at least qualitative, alignment with the aforementioned nonlinear optical measurements.

Finally, we turn to the thermal response of the structurally-evolving heterostructure. As thermal transport is a direct result of lattice vibrations, one would anticipate a strong correlation between the results demonstrated in our manuscript and the thermophysical properties of the superlattice. Indeed, as described in our prior work (Ravichandran et al. Nat. Mat. 2014), a trend in the thermal conductivity of the SLs that is inconsistent with incoherent, diffusive phonon propagation is found to exist in these superlattices below a critical periodicity. This anomalous trend in the thermal transport coefficient is ubiquitous evidence of a crossover from incoherent to coherent-like transport of phonons. However, the origin of this crossover had not been elucidated in prior studies – the increase in thermal conductivity associated with coherence is ultimately a direct result of the interfacial structural coupling demonstrated in this work. In other words, the structure-property relationship between coherent phonon transport (and ergo thermal conductivity) remained lacking in oxide superlattices to-date. This work provides direct evidence for interfacial coupling to be the underlying mechanism for such an observation of incoherent-to-coherent transport.

To provide a more direct measure of phonon transport, rather than observations in thermal conductivity, we have performed an additional set of experiments; namely, time-domain Brillouin scattering (TDBS) measurements. These ultrafast pump-probe experiments detailed in our revised manuscript provide a direct measure on the lifetime of longitudinal zone-center phonons in the oxide SL. Previous theory suggests that phonon *group velocities* increase greatly with reduced layer thickness, due to reduced zone folding, is largely responsible for the thermal conductivity trends in superlattices. However, our experiments verify that the observed trends in thermal conductivity are due to the phonon *scattering times*— these changes in vibrational scattering times are a direct result of the evolving superlattice structure and vibrational spectrum investigated here via high-resolution electron microscopy and spectroscopy. In other words, through a number of experimental approaches, we verify a direct correlation between the vibrational transport properties and the interfacial coupling within the atomistic structure.

In summary, the reviewer’s astute suggestions led us to perform an array of additional experiments to provide a more detailed narrative on the structure-property relationships of the unique structures investigated in our original submission. The results of these measurements, individually detailed in the above paragraphs and in our revised manuscript, strongly support a direct relationship between the observed interfacial coupling regimes of low-periodicity superlattices and the over-arching electronic, optical, and thermal properties of the heterostructure.

Regarding electronic properties of the superlattices, our revised text now reads on pages 8-9 as:

“

Figure 2. **Second-harmonic intensity indicates short-period superlattices lack interfaces.** Second-harmonic intensity of STO-CTO superlattices with varying periodicity, demonstrating various regimes of structural transitions and their role on electronic/optical properties of heterostructures. The error bars are calculated from the mean square deviation of a parabolic fit to the measured second-harmonic intensity vs incident electric field. Ball-and-stick models are included to pictorially show the connection between octahedral tilt and the presence structurally diffuse interfaces, or lack thereof.

We perform SHG to provide further evidence of the structural regimes, which also lends insight to the electronic properties of the superlattices, as shown in Figure 2. In contrast to linear optical measurements, which are dictated by the average of the linear response of the materials comprising the superlattice⁴⁶, the higher rank dielectric tensor associated with SHG vanishes if the constituent materials have inversion symmetry. Both STO and CTO have inversion symmetry, so the SHG intensity of STO-CTO superlattices is directly related to the polarizability of the interface where inversion symmetry is broken. An increase in SHG intensity is observed from SL27 to SL6 as the density of interfaces increases and layers remain independent with respect to each other. However, a marked decrease in the nonlinear optical response is observed in SL4 and rapidly vanishes in superlattices with short periodicities; as the heterostructure transitions from independent monolithic layers, to coupled layers, to a single centrosymmetric structure, the second-order optical response approaches zero. In other words, the structural transitions are directly reflected in the polarizability of the superlattices.”

Regarding the thermal properties of the superlattices, our revised text now includes a modified Figure on page 13 and text on pages 12 and 14:

“The predominance of these interface vibrations and their global response has been shown to affect the thermal characteristics of STO-CTO superlattices, where a crossover from incoherent to coherent phonon transport is observed as the heterostructure periodicity decreases.¹⁰ Previous reports have suggested that reduced zone-folding of phonon dispersion leads to an increased group velocity, but direct evidence connecting underlying phononic processes, new modes, and structure to the macroscopic transport mechanisms remains lacking.^{10,48}”

“

Figure 4: FTIR and time-domain Brillouin scattering response of STO-CTO superlattices. (a) Raw (solid), fitted (dashed), and residual (dot-dashed) data for UV-Raman from superlattices on an NGO substrate. 200 nm STO and CTO thin-films on substrates used to fit superlattice spectra are shown as the bottom of the panel. Difference curves are scaled by a factor of two for clarity. (b) Phonon lifetimes (black squares) as measured via time-domain Brillouin scattering in comparison to the thermal conductivity (red triangles, from reference 10) of STO-CTO superlattices with varying periodicities. The strong correlation between the two techniques conclusively demonstrates a transition in phonon scattering rates across the structural transitions elucidated upon with STEM/EELS. The error bars represent the standard deviation.

”

“To directly investigate the emergent phonon dynamics, we perform time-domain Brillouin scattering (TDBS) measurements, which detect the propagation of zone-center longitudinal modes as a function of time, thus providing a measure of their lifetime.^{53,54} For short-periods (SL2), an increase in both the phonon scattering time and thermal conductivity is observed; this is no longer described by a superlattice with interfaces, but rather has a single structure with uniform octahedral tilts. We find the thermal conductivity trends reported previously to be strongly correlated to the phonon scattering times (Figure 4), with both decreasing with increasing interface density to a minima at SL4.¹⁰ Thus, in strong agreement with both our high-resolution TEM, EELS, and SHG results, the

phonon lifetime is also found to increase as material interfaces vanish. This combination of the structural, electronic/optical, and vibrational characterization techniques unambiguously demonstrates the underlying coupling of heterostructures and emergent global properties that are driven by interfaces.”

We additionally include new DFT experiments in Supplementary information Section S3 that show the change in the Infrared active phonon DOS in SL4 and SL2 compared to bulk STO and CTO. The new Supplemental information also includes a visualization showing the eigenvectors of particular vibrational displacements that are unique to the superlattices and relevant to the experimental spectroscopies. The new supplemental Figure S8 and discussion are as follows:

“

Figure S8. Infrared active phonon DOS and eigenvectors showing emergent optically active vibrational response in superlattices. (a) IR-phonon DOS SL4 and SL2 showing the emergence of IR active high-energy vibrations that are not active in either STO or CTO. Eigenvectors of vibrational displacements in (b) SL2 and (c) SL4 showing layer and interface IR active modes.

As a demonstration of the impact of the present work on understanding emergent properties that derive from structural and vibrational features of the SLs, we show in Figure(a) the predicted IR-active-phonon DOS for the bulk STO and CTO and the IR-active-phonon DOS for SL2 and SL4. The IR-active-phonon DOS can be compared to the experimental FTIR spectra shown in Figure, Figure S,

Error! Reference source not found., and Error! Reference source not found.. We note emergent IR-active phonon modes at surprisingly high energies around 100 meV and also at very low energies. The emergent IR-active phonon modes consist of Ti-O modes localized within STO and CTO layers, delocalized between STO+CTO layers, and localized to interfaces. The density of the latter increases with increasing layer length. These emergent phonon modes are likely to underpin emergent properties, e.g., IR absorption and Raman spectra. Since in complex oxides, structure (e.g., octahedral tilts) and phonons are strongly coupled to electronic and magnetic properties, knowledge of the emergent phonons would help engineer novel properties. In particular, the displacement vectors of the high-energy modes have intriguing localization properties, either at the interfacial TiO₂ planes or within the layers that can potentially host unique spin structures.”

Regarding the infrared optical properties of the superlattices, our revised text now reads on pages 13-14 as:

“To connect the localized interface modes and structural transitions observed above to a macroscopic response, we perform Fourier Transform-Infrared spectroscopy (FTIR) (Figure 4(a)).^{10,15–18} The residuals from a fitted linear combination of STO and CTO films are quantified to accentuate changes that are unique to the superlattice. Due to selection rules, the spectra of STO contains less reflectivity minima relative to CTO. As the STO and interface incorporate CTO tilt patterns further loss of reflectivity should occur.^{49–52} Residual responses are observed near 500 cm⁻¹ and 560 cm⁻¹ that decrease with decreasing period thickness (Figure 4(a)). UV-Raman experiments show similar trends and are discussed in greater depth in the SI section S5. DFT informs that IR-active phonons emerge in the superlattices and are associated with layer-localized, layer-delocalized, and interface-localized Ti-O vibrational modes that are not IR-active in bulk STO or CTO, as shown in Figure S8. The residual responses are similar in energy to those observed locally with both EELS and DFT. Additionally, the sum of residuals (Figure S16) scales with interface density. We therefore conclude they are a consequence of the local displacements existing at the interfaces of the superlattice.”

In addition to the new experiments directly connecting our observations to macroscopic properties, we have included additional references to electronic phenomena in ordered perovskites on page 2:

- Balachandran, Prasanna V. “Massive Band Gap Variation in Layered Oxides through Cation Ordering.” *Nature Communications*, 2015, 7.
Here, Balachandran show that chemically ordering a Ruddlesden-Popper oxide unit-cell, which is closely related to perovskite oxides, can vary the band gap of the material from ~1-3 eV.
- Mulder, Andrew T., Nicole A. Benedek, James M. Rondinelli, and Craig J. Fennie. “Turning ABO₃ Antiferroelectrics into Ferroelectrics: Design Rules for Practical Rotation-Driven Ferroelectricity in Double Perovskites and A₃B₂O₇ Ruddlesden-Popper Compounds.” *Advanced Functional Materials*, May 6, 2013, 4810–20. <https://doi.org/10.1002/adfm.201300210>.
Here, Mulder et al. show that chemical ordering in a perovskite oxide leads to emergent ferroelectricity.
- Rondinelli, James M., and Craig J. Fennie. “Octahedral Rotation-Induced Ferroelectricity in Cation Ordered Perovskites.” *Advanced Materials* 24, no. 15 (April 17, 2012): 1961–68. <https://doi.org/10.1002/adma.201104674>.

Here, Mulder et al. also show that chemical ordering in a perovskite oxide leads to emergent ferroelectricity.

The authors have also pointed toward interface driven emergent properties in perovskite superlattices on page 7:

- Changed “..., as occurs for in-phase tilt angles in other dissimilar perovskite tilt-systems.^{3,16–21}” to “Similar coupling regions observed in other perovskite heterostructures result in extraordinary electrical and magnetic properties.^{3,7–9,19,19–24,41}” to emphasize the impact of octahedral coupling in a broader set of material properties.
- The authors have added three new references that refer to electronic and magnetic properties:
 - Zhai et al., “Correlating Interfacial Octahedral Rotations with Magnetism in $(\text{LaMnO}_{3+\delta})_N/(\text{SrTiO}_3)_N$ Superlattices”
 - Hwang et al., “Structural Origins of the Properties of Rare Earth Nickelate Superlattices”
 - Chen et al., “Spatially Controlled Octahedral Rotations and Metal–Insulator Transitions in Nickelate Superlattices.”

Specific comments :

- Page 2: The first sentence this should be toned down as this is known for strained single thin films

We appreciate the Reviewer’s suggestion and agree that alternate phrasing would both increase clarity to potential readers as well as more accurately contextualize our statement. On page 2 of our manuscript, we have revised the sentence

“However, the presence of interfacial vibrations that impact phonon-mediated responses, like thermal conductivity^{7,8}, has only been inferred in experiments indirectly.”

to

“However, the interfacial vibrations that impact phonon-mediated properties, like thermal conductivity^{10,11}, are measured using macroscopic techniques that lack spatial resolution.”.

- Page 2: Wording “Symmetries atypical of either constituent” is not clear

To increase clarity to potential readers, the revised text on page 2 has been changed from

“Symmetries atypical of either constituent material are observed within a few atomic planes near the interface.”

to

“Structurally diffuse interfaces are observed that bridge the bounding materials.”

- Page 2: I suggest toning down the sentences between materials and interfaces, which are known to the community.

We appreciate the Reviewer's suggestion, and have toned down the statement accordingly. The revised text can be found on page 2 of our manuscript and now reads as:

"Our results provide direct visualization of the progression of the local atomic structure and interface vibrations as they come to determine the vibrational response of an entire superlattice."

- Page 2: precise what you mean by "spatial resolution". Lateral and or depth?

By "spatial resolution" we are referring to spatial resolution in general, as we are concerned with local signal from the interface. All (S)TEM measurements have interaction through the entire projected depth of the specimen as the high-energy electron transmits through the material. The projection is the reason we refer to "atomic columns" rather than individual atoms. Even with that being the case, our TEM samples are cross-sections, and when the probe is near the interface the electron is always traveling along the interface, through the TEM sample. This is inherent to the geometry of the cross-sectional samples. With this cross-sectional geometry our electron is always sampling a local interface signal when our probe is near the interface. The STEM probe and signal is also very local with small radial sampling. We use off-axis electron energy-loss spectroscopy to make sure that the vibrational signal we acquire is laterally local. The electron-vibration interactions are complicated and beyond the scope of this paper.

This aspect of our experimental design was perhaps unclear in our original text. To increase clarity to potential readers, our revised manuscript defines the first instance of "spatial resolution" on page 2 to read as "lateral spatial resolution" (see Refs. 31 and 33 for discussion on off-axis versus on-axis vibrational EELS)

- Page 3 and following parts: I suggest describing the potential limitations of DFT calculations for the phonon response of the investigated nano-features. Has the interface a defined symmetry?

DFT is a ground-state theory and, therefore, does a better job for phonons than electronic excitations. The results depend on the exchange-correlation functional. We adopted the local density approximation (LDA) because it has been found that it does better for phonons in bulk CTO and STO at the Γ point, which is of interest here. We have inserted this point in the DFT methods section (Section 5.3) of the main manuscript as follows:

"Phonon calculations were performed using the LDA for exchange-correlation because it has been found to do better for phonons at the Γ -point, which is of interest here, in bulk CTO and STO.^{51,60,61"}

DFT phonon calculations often produce modes with imaginary frequencies, which normally suggest instabilities. No imaginary-frequency modes appear in SL2 and SL4, but they do appear in SL8 phonon calculations. They are likely caused by the fact that certain strain relaxations cannot be included while maintaining the integrity of the supercells. In the Methods section, we explain how we handled the phonon calculations for SL8.

The references 51, 60, and 61 in the new sentence provided above are as follows:

51. Himmetoglu, Burak, Anderson Janotti, Hartwin Peelaers, Audrius Alkauskas, and Chris G. Van de Walle. "First-Principles Study of the Mobility of SrTiO₃." *Physical Review B* 90, no. 24 (December 23, 2014): 241204. <https://doi.org/10.1103/PhysRevB.90.241204>.
60. Železný, V., Eric Cockayne, J. Petzelt, M. F. Limonov, D. E. Usvyat, V. V. Lemanov, and A. A. Volkov. "Temperature Dependence of Infrared-Active Phonons in CaTiO₃ : A Combined Spectroscopic and First-Principles Study." *Physical Review B* 66, no. 22 (December 17, 2002): 224303. <https://doi.org/10.1103/PhysRevB.66.224303>.
61. Zhang, Yubo, Jianwei Sun, John P. Perdew, and Xifan Wu. "Comparative First-Principles Studies of Prototypical Ferroelectric Materials by LDA, GGA, and SCAN Meta-GGA." *Physical Review B* 96, no. 3 (July 24, 2017): 035143. <https://doi.org/10.1103/PhysRevB.96.035143>.

- Page 3: What is meant by "length scale of octahedral coupling", please explain.

We thank the Reviewer for this astute question, as "spatial resolution" is an important aspect of our work that we would like to convey to the scientific community at-large. Specifically, the authors are referring to the width of the structurally diffuse interfaces in comparison to the width of the layers. To increase clarity to potential readers, we have revised the text on page 4 from

"We show that as the superlattice layer thickness approaches the length scale of octahedral coupling the layers lose uniqueness and adopt the symmetry and vibrational response of the interface."

to

"We show that as the superlattice layer thickness approaches the width of structurally diffuse interfaces, where octahedral coupling occurs, the layers lose uniqueness and adopt the structure and vibrational response of the interface."

- Page 4 and following: Please be precise what unit cell means (the pseudo-cubic unit cell or the tilt-induced cell-doubling unit cell)

To increase clarity to potential readers, we have revised our text on page 4 to directly ref to pseudo-cubic unit-cells and *not* the orthorhombic unit-cell for describing the periodicity of the superlattices. This revised text on page 4 now reads as:

"To evaluate the influence of interfaces, we synthesized three STO-CTO superlattices featuring layer thicknesses of 27, 6, 4, 3, and 2 pseudo-cubic unit-cells (SL27, SL6, SL4, SL3, SL2), as shown schematically in Figure 1(a)."

- Page 4 & 5: Would it be possible to estimate the octahedra tilt angle from the intensity of the superstructure reflections?

This an astute comment from Reviewer 1 who clearly has knowledge of the interplay between structural ordering and diffraction. In a perfect world we could quantify octahedral tilt angles using the ordered and fundamental Bragg reflection intensities. The work “Woodward and Reaney, “Electron Diffraction of Tilted Perovskites.” includes the theory for a perfect (single scattering, kinematic) world. Unfortunately, electrons undergo multiple scattering (we refer as dynamic scattering) that allows for electrons to scatter from one Bragg reflection into another. The dynamic scattering makes relating intensity to structure factor very complicated and introduces thickness dependence into intensity measurements. To accurately understand the connection in reciprocal space one would need to perform convergent-beam electron diffraction accompanied with multi-slice simulations for comparison. The apertures used for our selected-area electron diffraction experiments are also 60+ nm wide and would provide an average representation of octahedral tilts present in all selected layers and interfaces. The authors briefly mention that we do not pursue quantifying tilt-angles using the diffracted intensities in the Supplemental information surrounding the 4D-STEM (scanning convergent-beam diffraction) experiments. In the end, the authors decided to pursue real-space quantification because we can directly measure the tilt-angles and asses how they spatially change.

An interesting future study would be quantifying the tilt-angle as a function of position using 4D-STEM. This is currently attracting attention, see Nord et al., “Atomic Resolution HOLZ-STEM Imaging of Atom Position Modulation in Oxide Heterostructures.”. With detailed simulations that account for tilt-angles and multiple scattering, one could conceivably quantify octahedral tilt-angles around each of the pseudo-cubic axes at each real space dimension of the four-dimensional (x, y, q_x, q_y) dataset. We again thank Reviewer 1 for this comment and hope that we addressed their question. The authors may pursue the suggestion in future research given the increasing interest in 4D-STEM and local characterization of octahedral tilt patterns.

- Page 4 & 5: Can you please say something about potential diffuse scattering which could indicate the loss of translation symmetry or changes in the coherence. Is the width of the superstructure reflection the same for different families?

This is a very keen observation of Reviewer 1. In the lead author’s dissertation (now cited as reference 67) , details on the diffuse background present in the selected-area diffraction patterns were discussed in-depth. In our original submission, we did not include this level of depth due to length considerations. However, we have extended the supplemental information discussion surrounding the selected-area diffraction patterns to include such discussion on the long-range coherence of the diffracting wave as discussed below. To directly answer the reviewer’s astute question: Yes, there is diffuse scattering and/or peak broadening associated with the number of diffracting layers vs atomic lattice.

To expand on this exciting future avenue of diffuse scattering in nanoscale heterostructures to potential readers, the authors have included additional text in our revised manuscript on page S5-6. This additional text reads as:

“In the SADPs shown in “(b-d) a rich set of information is present between the Bragg peaks. In SL27 shown in “(b), closely spaced superlattice reflections extend from the Bragg peaks making them look

streaked. The same superlattice reflection are seen in SL4 and SL2, but are much further spaced and appear as distinct peaks because of the much larger real-space periodicities. An interesting diffuse background intensity is found in SL4 at the location of the ordered reflections. We will speculate to their origin but defer to future research to ascertain the true origin of the diffuse intensity. Diffuse intensity in the background of diffraction patterns is associated with the loss of long range coherence, whether it be from structural static disorder or thermal vibrations. In the present case, we find that the superlattices contain a high degree of long-range static ordering, as clear from the sharp fundamental, ordered, and superlattice Bragg reflections. This remaining option is thermal vibrations. In many phase transitions a phonon mode at a specific momentum vector causes the transition to a lower symmetry structure. In these cases the thermal diffuse scattering, typically considered as a uniform Gaussian like distribution centered at $\mathbf{q}=(000)$, can become non-uniform and slowly develop into Bragg peaks as the mode softens and forms new Brillouin zones.^{49–51} In the case of SL4 we observe with iDPC that octahedral tilting is present throughout the entire superlattice structure, including within STO layers where no natural tilts are present at ambient conditions. SL4 also has the smallest number of unit-cells per layer where the layers can be well defined, see section **Error! Reference source not found.** and the main text for elaboration. From the continuous change in tilting, it is expected that the phonon associated with the tilting is neither completely hardened or softened through the entirety of the structure and may therefore produce a non-uniform thermal diffuse background.”

- Page 5: “the orientations of octahedra tilts are different”. What’s about the amplitude of the tilts?

In short, yes, the amplitudes are different. The authors have addressed the amplitude of the tilt angle directly after the selected-area diffraction discussion in the imaging section. The magnitude of the tilt-angles are shown in Figure 1(g,j,m) and further discussion and quantification can be found in the supplemental information section 2 (page S9), which is referenced in the main text. The supplemental information section that details the numerical quantification of tilt magnitude reads as follows:

“In the STO layer of SL27 an in-phase tilt-angle of 1.84° and 2.69° STO was measured in the TiO₂ and AO planes, respectively, since no tilting is present and represents a bound to measurement error. In CTO out-of-phase tilt-angles of 10.04° and 10.32° in the TiO₂ and AO planes, respectively. In SL4 the quantified out-of-phase tilt-angle profile was sinusoidal with an average of 6.34° and 7.39° in the TiO₂ and AO planes, respectively, demonstrating that the layers tilts have accommodated to approach the mean or interface value observed in the larger-period SL27. In SL2 the tilt profile did not have any systematic tilt-angle related to the layer periodicity appearing “dephased” with a nearly constant tilt-angles of 7.138° and 6.096° in the AO and TiO₂ planes, respectively.”

- Page 5 “single crystal pattern”. What does this mean? What about the FWHM of the reflections?

The authors could have made the terminology clearer. Specifically, when we refer to “single crystal pattern”, we are referring to the “experimental pattern” observed in our SL structures and it’s strong resemblance to that of a “single crystal diffraction pattern”, i.e. single diffraction pattern with single orientation and lattice parameter. If one could place a 20 nm aperture in a 21 nm grain, then the diffraction pattern would look like a single crystal with some minimal broadening from the shape

function of the diffracted wave attributed to the aperture limited coherence. In a TEM, unlike XRD, the FWHM of a peak is strongly impacted by other experimental parameters (ex. probe convergence angle) and is not as direct of a measure of polycrystallinity as simply identifying the presence or Bragg peaks (or rings) in the 2D diffraction pattern. For example, in SL27 there are Bragg reflections from two $Pnma$ orientations. In SL4 and 2 there are only Bragg reflections from one $Pnma$ orientation, which we conclude is diffraction from one single crystal. In our case the 180 nm selected-area aperture spans nearly the entire film thickness of the 200 nm thin-films and is a secondary contribution to the shape function, with convergence angle probably determining the width of Bragg peaks.

To increase clarity to potential readers on this topic. and avoid any potential inconsistencies, we have revised our manuscript to eliminate these terms and use a more direct phrasing. Specifically, our revised manuscript on page 6 has been revised from

“Thus, as the layer thickness decreases, the underlying crystal structure adapts, resembling a single crystal pattern, which could be enabled by tilting of the TiO_6 octahedra.”^{3,4,19,21–24”}

to

“Thus, as the layer thickness decreases, the underlying crystal structure adapts, ~~resembling a single crystal pattern,~~ which could be enabled by tilting of the TiO_6 octahedra.”^{3,4,19,21–24”}

- Page 6 “Fm-3m” : do you mean the simple cubic structure “Pm-3m” ? Or do you infer chemical ordering?

The authors thank Reviewer 1 for noting this typographical error. Yes, we intended to state $Pm\bar{3}m$ and not $Fm\bar{3}m$. This error, on page 7 of our revised manuscript, has been corrected.

- Page 7: The term “crystal of interfaces” is unusual and likely not correct. Same for “crystalline hybrid” on page 10. Two unusual terms for the same object. Since the authors address phonons, it could be useful to reflect on the two proto-type behaviors of mixed systems: virtual ion behavior and two-mode behavior. What is the observed regime?

We appreciate the Reviewer’s comment and suggestion regarding this unusual phrasing. However, we respectfully disagree on its correctness and applicability in our current manuscript. Our rationale behind this specific phrasing of the observed SL2 structure is to convey a qualitative depiction of the atomistic structure to a wide-audience of readers; we strongly believe that this terminology aides in visualization of the origin of the structural distortion and vibrations in the extreme limit of coupling between STO and CTO unit-cells across their interface. In this extreme limit, the previously-independent layers couple and form a single homogeneous structure, where the repeating unit mimics that of the *interface* that existed between the two layers. In other words, as the crystal transitions, it takes a form that is akin to that of solely interfaces, with individual ‘layers’ being ill-defined.

Perhaps more specifically, if *any* basis is repeated on a lattice, then we define it as a crystal. In this work, we find that the atomic arrangement, or tilts, found at the interface of SL27 or SL4 is repeated

everywhere uniformly in SL2. The structure of SL2 is therefore a crystal derivative of interfaces in the large period structure.

While the reviewer is certainly correct in that, technically, bounding materials are required to define an interface, we believe the phrasing illustrates a more apt-depiction of the unique structure elucidated on in this work. To enable clarity to potential readers from the scientific community at-large, we believe an italicized format for the current phrasing is a strong middle-ground between qualitative illustration and the more rigorous materials science definition.

Regarding various models: the authors agree that a discussion of such models is certainly appropriate and would strengthen our manuscript. However, a discussion adequately addressing the suggested models would be restrictively long. Therefore, in our revised manuscript, we have added a discussion on the virtual crystal model to the Supplemental information and added reference to such in the main text so that the discussion so that we suitably compare our findings in light of the prior existing model. This additional text found in the SI section S4, pages 21-22, can be found below for the reviewer's convenience.

"It is useful to assess the current observations with regard to models for mixed systems, such as the virtual crystal model or two-ion model.^{48,72} In the virtual crystal model structural or chemical heterogeneity at interface are incorporated into an interface layer, much like presented in the current work.⁷² However, in the virtual crystal model, the interface layer is assigned properties that are effectively represent a disordered combination of the bounding materials. In the present case the interface is not disordered and contains vibrations that cannot be explained by the bounding materials. The structurally diffuse interface cannot be captured by a virtual crystal because of its unique structural and vibrational state. In a two-phonon model of superlattices phonons are grouped into either coherent or incoherent phonons.⁴⁸ The coherent phonons with long wavelengths and mean-free-paths can propagate through the material unscattered, while the incoherent phonons have short wavelengths and mean-free-paths and scatter from the interfaces. In SL27, the two-phonon model may adequately describe the transport of the material with the exception that the scattering probability of some incoherent phonons may not be as large as in a structurally and chemically abrupt interface because the interface structure and vibrations can mediate the transition from one layer to the other. In SL4, the incoherent phonons will also scatter form the interfaces, and there is a much higher density of interfaces so there will be a larger accumulation of scattering probability across the thickness of the superlattice. However, the incorporation of tilt into the STO layers makes the phonon modes more like the CTO modes such that transmission might be increased. In short-period case of SL2, the concept of a two-phonon model need not be considered. The structure in effect no longer has interfaces and the vibrational structure of the entire superlattice is uniform."

- Page 7 and others: A reminder of how crystal structures vary in the STO-CTO solid solution could be insightful.

Reviewer 1 is correct that a reminder of STO-CTO phases would be insightful. However, due to length considerations and extensive review of such topics in literature, we do not believe it provides enough essential information to be added to the primary manuscript.

- Page 8: The interpretation of the DOS is at the hard limit of overinterpretation. For instance a peak shift does not formally indicate a change in symmetry, but could also be a shift of atoms within a given space group, as known for so-called soft modes.

We thank the reviewer for raising this issue. We unfortunately misspoke. We did not intend to convey the notion that changes in the space group control phonon shifts. We are in fact referring to octahedral tilts, which may or may not correlate with space-group changes. We have reworded the relevant text to convey the correct message that it is changes in octahedral tilts that correlate with the phonon-frequency shifts. The text now reads as follows:

“Since phonon frequencies are affected by changes of bond lengths and bond angles, we expect the evolution of the octahedral tilts in our superlattices (Figure 1) to affect the phonon density of states (DOS), which drives the inherent thermal and infrared optical properties.”

- Page 9, Figure 2(k). I guess we should read here SL2 and not SL4

The authors thank Reviewer 1 for noting this typographical error. The reviewer is correct: we intended to state ‘SL2’ rather than ‘SL4’. This error is corrected in the revised manuscript, and the revised Figure is shown below for the reviewer’s convenience. While discussing the below figure, the authors would also like to not that we have substituted panels (f-h) with a higher spatial resolution dataset that was recently acquired. The spatial higher resolution scan increases the clarity of small spectral changes in (g) but altogether exhibits the same behavior and trends in (f-h).

- Page 11: The question “Do the interface vibrations ..” is trivial and the answer “Yes” is known and accepted in the community.

We appreciate the reviewer’s suggestion and agree with their associated comments. To enhance the narrative for potential readers, we have modified this section of the manuscript to discuss the various structure-property relationships noted in a previous response above. In the revised manuscript the question has been removed entirely.

- Page 11, Figure 3. Title: “Raman” should be added as not only IR data are shown. The reviewer is correct, and we greatly appreciate the suggestion. We have corrected this in our revised manuscript, and the caption includes the Raman addition in the title sentence in addition to the thickness of the thin-films. The original Figure 3 has also been moved to the supplemental information in place of a new Figure. The full caption of Figure S12 now reads:

“Figure S12. Interface driven modifications in the macroscopic Raman and infrared response of STO-CTO superlattices. Raw (solid), fitted (dashed), and residual (dot-dashed) data for (a) UV-Raman and (b) FTIR spectra taken from superlattices on an NGO substrate, and (c) FTIR spectra taken from superlattices on a LSAT substrate. 200 nm STO and CTO thin-films on substrates used to fit superlattice spectra are shown as the bottom of each panel. Difference curves are scaled by a factor of two for clarity.”

I suggest adding a reference spectrum of NGO to support the statement that the substrate is not seen by Raman scattering, same for IR.

The STO and CTO standards are also on NGO substrates. The lack of first-order response in the STO indicates scattering from only STO. To enhance clarity to potential readers, we demonstrate that the current measurements do not include contributions from the NGO substrate by adding an additional supplemental information figure that indicates where first-order Raman active NGO modes exist. The following figure has been included in the Supplemental information in Section S5 as Figure S13.

Figure S13. UV-Raman acquired from thin-films on NGO substrates. Grey dotted lines in (e) and (f) indicate energies where first-order NGO Raman peaks would appear if the substrate was being sampled.

Title of (c) should read “FTIR LSAT”.

The reviewer is correct that this is a typographical error. This comment is referring to our old Figure 3, which has been moved to supplemental information section S5 in place of a new figure. The revised figure is shown below.

Figure S12. Interface driven modifications in the macroscopic Raman and infrared response of STO-CTO superlattices. Raw (solid), fitted (dashed), and residual (dot-dashed) data for (a) UV-Raman and (b) FTIR spectra taken from superlattices on an NGO substrate, and (c) FTIR spectra taken from superlattices on a LSAT substrate. 200 nm STO and CTO thin-films on substrates used to fit superlattice spectra are shown as the bottom of each panel. Difference curves are scaled by a factor of two for clarity.

An example of the raw, fitted, residual data should be given in more detail and with a zoom in the supporting information material to allow a proper and critical inspection, which is not possible with the current Figure 3. Only with this a referee or author can further judge on the described analysis and interpretation.

The authors thank Reviewer 1 for the recommendation. The authors have added a figure in the Supplemental information for each of the Raman and FTIR cascades, each with a larger dedicated panel for each superlattice. The three supplemental information figures can be found below for Reviewer 1's convenience.

Figure S13. UV-Raman acquired from thin-films on NGO substrates. Grey dotted lines in (e) and (f) indicate energies where first-order NGO Raman peaks would appear if the substrate was being sampled.

Figure S14. . FTIR acquired from thin-films on NGO.

Figure S15. FTIR acquired from thin-films on LSAT.

- Page 11: The “emergent Raman response” near 220 and 500 cm^{-1} should be interpreted in terms of the expected vibrational pattern expected in the respective spectral region (tilt, cation displacement, breathing etc.). Between 200 and 500 cm^{-1} the overall second order Raman signature is much stronger, this might mask other features.

Reviewer 1 makes a good suggestion here. From our data, one cannot directly assign modes directly associated with Ti- and O- vibrations without possible overinterpretation. However, we note a number of prior works that have accurately assign spectral regions to O-Ti-O bending vibrations and Ti-O₃ torsional modes. To address this aspect of our revised manuscript, we have included additional text to our Supplemental information regarding the discussion of our UV-Raman data. This additional text, found in Section S5 (page 24), now reads as:

“Features in the region between 200-400 cm^{-1} have previously been interpreted,^{49,62,65} for CTO, as resulting from O-Ti-O bending vibrations while features in the region between 400-600 have been assigned as Ti-O₃ torsional modes.^{49,62,65} Due to the large mass of Ca or Sr atoms, vibrational modes involving their motion are not expected to be observed in the region between 200-500 cm^{-1} , but rather would occur at lower frequencies not observed by our experiments. Thus, although we cannot make a concrete assignment to each emergent Raman feature, we can infer that spectral features in this region reflect properties of the Ti-O sublattice.”

- Page 16/17: The UV light is thus very differently absorbed by STO and CTO, thus the response for a given small volume must be very (!) different is the described decomposition of the spectra under this condition still stable. Please give the thickness of the investigated monolithic films.

To ensure consistency with our other measurements, the STO and CTO monolithic films have an identical thickness of 200 nm; this important aspect of our measurements is now noted in the revised manuscript on Section S5 (page 21). Further, we note this feature within the caption of Figure S12, which contains the pertinent UV-Raman results on STO/CTO monolithic films.

We further note that Reviewer 1 is absolutely correct in their statement about the absorption of the two different materials. Specifically, we note that at the wavelength used for our UV-Raman measurements, the skin-depth is approximately 26 nm within STO, while being in excess of 1 micron for CTO. However, despite the transparency of CTO, all Raman films examined were identical in thickness (200 nm), and therefore contain 100 nm of STO; as this is more than three times the skin-depth, the underlying NGO substrate should not affect the Raman measurement in any of the SLs investigated here. To increase clarity on this important feature of our measurements to potential readers, we have added the following statement to the Methods section of our revised manuscript. This text on Section S5 (page 19) reads as:

“At this wavelength, the skin depth for the exciting UV-light is 26 nm within STO while being $>1 \mu\text{m}$ for CTO. Despite the transparency of CTO, all Raman examined films were 200 nm in thickness and thus contain at least 100 nm of STO. This is more than three times the skin depth and thus the underlying NGO does not impact the Raman experiment.”

2 Reviewer #2 (Remarks to the Author):

In the manuscript “Nanoscale Phonon Spectroscopy Reveals Emergent Interface Vibrational Structure of Superlattices”, Hoglund et al. use imaging techniques and electron energy loss spectroscopy in the (scanning) transmission electron microscope to show evidence for an emergent vibrational response in oxide superlattices. The observation that, with decreasing superlattice layer thickness, the local vibrational response converges towards that of the layer interfaces, is attributed to changes in atomic scale symmetry uncharacteristic of either “bulk” constituent material. Local microscopy measurements are supported by theoretical modelling and global measurements using Raman and FTIR optical spectroscopies. The results are overall convincing and support the authors conclusions. Key to this work is the combination imaging and spectroscopy techniques, providing novel insight into emergent vibrational properties of solids down to the atomic scale. These results are likely of great interest to a wide audience.

However, before I can recommend this manuscript for publication, the following points should be addressed by the authors.

While the data presentation is good overall, the plot legends for the black-line plots in figures 2(k) and S10(f) be “SL2” not “SL4”. Moreover, should the plot title in fig. 3(c) should be “FTIR LSAT” and not “LSAT NGO”?

Reviewer 2 is correct that this is a typographical error. The authors had intended to have SL2 and not SL4; we have corrected the text in our revised manuscript. The updated Figure is shown below and can be found on page 12 of our revised manuscript. While discussing the below figure, the authors would also like to note that we have substituted panels (f-h) with a higher spatial resolution dataset

that was recently acquired. The spatial higher resolution scan increases the clarity of small spectral changes in (g) but altogether exhibits the same behavior and trends in (f-h).

Figure 1. Localized vibrational response of superlattices indicates emergent role of the interfacial symmetry. (a) DFT-calculated phonon DOS of SL27, SL4, and SL2 models. Arrows indicate the dominant phonon peaks. Cascade of (b) DFT-calculated phonon DOS projected on STO (green), CTO (purple), and interface (orange) layers and the total DOS (black) each for superlattice model. (c-k) Monochromated STEM-EELS line profile analyses of the three SL structures: (c-e) SL27, (f-h) SL4, and (i-k) SL2 each with the (c,f,i) ADF profile, (d,g,j) EELS profile, and (e,h,k) integrated spectra from each layer (as indicated by colored regions in the ADF profile)

Figure 1: the authors use red, green, and blue “atoms” to label atomic column positions without clearly defining what these represent. Adding labels to the figure and/ or a description of this in the figure caption would be appropriate.

We appreciate Reviewer 2’s suggestion on increasing the clarity of this figure. To improve the clarity of Figure 1 the authors have included the atom colors in the caption, in addition to an inset within the Figure itself. We have included the revised Figure caption below for Reviewer 2’s convenience.

“Figure 1. **Period-dependent changes in the symmetry of STO-CTO superlattices.** (a) Superlattice structures calculated from DFT with colored-bar schematics denoting the (left) chemically and (right) structurally defined interfaces. Here green, blue, and cyan rectangles correspond to STO, CTO, and interface layers, respectively; the same colors are used in e,f,h,i,k, and l panels. Green, blue, grey, and red circles in a,e,f,h,i, and k correspond to Sr, Ca, Ti, and O atoms, respectively. [100] zone-axis SADP in (b) SL27, (c) SL4, and (d) SL2 grown on NdGaO₃. Colored arrows correspond to ordered reflections from the three possible domains. Solid arrows indicate ordered reflections that exist, and hollow arrows indicate absences. Insets in b-d show ball-and-stick models of the orientations present with border

colors matching the arrows. Red and blue arrows and insets are viewed along an out-of-phase tilt-axis and the yellow are viewed along an in-phase tilt-axis. In (c) and (d) superlattice reflections are seen in the 001 direction. In (b), closely spaced superlattice reflections appear as streaking of the fundamental reflections. (e,h,k) ADF, (f,i,l) iDPC images, and (g,j,m) octahedral tilt-angles of (e-g) SL27, (h-j) SL4, and (k-m) SL2. The legend inside (g) illustrates the (green) in-plane and (black) out-of-plane tilt-angles. The tilt-angles for a one unit-cell column are overlaid in each iDPC image to demonstrate the changing in-plane (green triangles) and out-of-plane (grey triangles) tilt-angles. In (g,j,m), solid and dashed curves are from experimental measurements and calculations, respectively. Error bars represent one standard deviation. Chemically abrupt interfaces are illustrated to the left of ADF images (e,h,k) and model structures (a), illustrating the abrupt change between STO (green) and CTO (blue) layers. Chemically diffuse interfaces are illustrated to the right of iDPC images (f,i,l) and model structures (a), illustrating the non-abrupt symmetry changes that are occurring as a result of octahedral coupling.”

On lines 225-227, the authors state: “...there are features in the interface spectra that cannot be reproduced by a mixture of the bulk-like CTO and STO phases (shown in more detail in supporting information Figure S10(a,b)).” As emergent properties are the focus of this manuscript, the authors should elaborate on this point.

We have expanded upon this comment in both our revised main text as well as the Supplemental information. Specifically, on page 12 of the main manuscript, the text now reads as:

“Furthermore, we note that, while the interface and total-structure spectra bear some similarities, there are features in the interface spectra that cannot be reproduced by a mixture of the bulk-like CTO and STO phases (shown in more detail in Figure S11(a,b)). Unique vibrations emerging from the octahedral coupling at the interface must be present to account for the discrepancy between the total and interface spectra, as calculated via DFT.”

As we agree that even further detail would greatly increase clarity to potential readers of our work, we have also expanded the discussion in the SI on page 20 (e.g., the section referenced in the main manuscript). The revised discussion now reads as:

“In SL27, the STO, CTO, and interface layers each have unique responses that agree with trends found in DFT experiments. DFT informs that the vibrations in the 30-120 meV energy range are from the Ti and O sublattices, and that the peaks in the Ti+O projected DOS shift with tilt-angle. We can conclude that the experimental observations are from layer-to-layer changes in TiO₆ tilt. To further emphasize this conclusion, the total and interface spectra can be compared. One would expect the interface spectra to match the total superlattice spectrum if unique interface vibrations are not present because both would be a linear combination of 50% STO and 50% CTO. Instead, there are discrepancies between the interface and total spectra that originate from vibration in the structurally diffuse interface. The small volume fraction of the region ascribed to the interface relative to the large STO and CTO layers suppresses the interface contribution to the total response. See section S2 for discussion regarding the volume fraction of layers. With the discrepancies, and agreement with DFT, we can conclude that the vibrational EELS experiments of SL27 is measuring local changes in vibrations at the interfaces that are a result of an octahedral coupling region. Furthermore, the interface and total spectra are nearly identical when the number of unit-cell per layer is reduced to

four, such that the volume fraction of interfaces is identical to the volume fraction of either STO or CTO layers. Now that the interface represents an appreciable portion of the material, the interface vibrational response emerges and contributes appreciably to the total response. The interface contribution to the total vibrations is in addition to the influence that the interface has on the tilts in the STO layers, which will make the STO vibrational response more like the interface response, as also described by DFT.

It is useful to assess the current observations with regard to models for mixed systems, such as the virtual crystal model or two-ion model.^{48,72} In the virtual crystal model structural or chemical heterogeneity at interface are incorporated into an interface layer, much like presented in the current work.⁷² However, in the virtual crystal model, the interface layer is assigned properties that are effectively represent a disordered combination of the bounding materials. In the present case the interface is not disordered and contains vibrations that cannot be explained by the bounding materials. The structurally diffuse interface cannot be captured by a virtual crystal because of its unique structural and vibrational state. In a two-phonon model of superlattices phonons are grouped into either coherent or incoherent phonons.⁴⁸ The coherent phonons with long wavelengths and mean-free-paths can propagate through the material unscattered, while the incoherent phonons have short wavelengths and mean-free-paths and scatter from the interfaces. In SL27, the two-phonon model may adequately describe the transport of the material with the exception that the scattering probability of some incoherent phonons may not be as large as in a structurally and chemically abrupt interface because the interface structure and vibrations can mediate the transition from one layer to the other. In SL4, the incoherent phonons will also scatter from the interfaces, and there is a much higher density of interfaces so there will be a larger accumulation of scattering probability across the thickness of the superlattice. However, the incorporation of tilt into the STO layers makes the phonon modes more like the CTO modes such that transmission might be increased. In short-period case of SL2, the concept of a two-phonon model need not be considered. The structure in effect no longer has interfaces and the vibrational structure of the entire superlattice is uniform.”

Figure 3: while the residual Raman spectra clearly show an increasing intensity with decreasing layer thickness, this is not so obvious for the IR measurements. It would be interesting to attempt to quantify this difference, helping the reader better grasp the magnitude of this effect. This would also help substantiate their claim that FTIR residual spectra show a “similar evolution” to that of the Raman residual spectra. While there may be many ways of going about this, one suggestion is to plot the summed absolute values over regions of interest and/ or the full range in the residual spectra shown in figure 3, as a function of layer thickness, for both Raman and IR.

We greatly appreciate the reviewer’s comment and accompanying suggestion. Before detailing our response and revisions, the authors would like to inform Reviewer 2 that the original Figure 3 has been moved to Supplemental information Section 5 as Figure S12 in place of a new Figure 3. The “(b) FTIR NGO” panel is included in the new Figure 3.

To increase clarity on the observed changes in our optical spectra with varying periodicity, we have incorporated the reviewer’s suggested figure to the Supplemental information. The figure and associated caption are shown below for convenience.

Figure S16. Sum of residuals above 400 cm^{-1} from Raman and FTIR experiments.

Additionally, as Reviewer 2 noted, the changes between the superlattice-to-superlattice responses are sometimes subtle, particularly due to their reduced dimensions (e.g., thin film response). To allow for the reader to more readily observe subtle features in the raw and fitted data we have included three additional supplemental information figures (Figure S13-S15), each of which have a dedicated panel for each superlattice with much larger plots. The three new supplemental information Figures S13-S15 are included below for Reviewers 2's convenience.

“Figure S13. UV-Raman acquired from thin-films on NGO substrates. Grey dotted lines in (e) and (f) indicate energies where first-order NGO Raman peaks would appear if the substrate was being sampled.

Figure S14. FTIR acquired from thin-films on NGO.

Figure S15.. FTIR acquired from thin-films on LSAT.”

In section 5.2, on line 311, the authors state that they “only integrate pixels in the top half of the acquired signal”. I assume here they are referring to the “top half” of the spectrometer camera, as displayed on the computer screen. It is not obvious from the text why the authors made this choice, thus a justification should be given.

We thank the reviewer for commenting on this subtle detail in our experimental technique. Our reasoning has now been included in Section 5.2, which is described by the following text:

“By integrating the signal at higher angles we preferentially select vibrations that have undergone impact scattering, which is a more spatially localized signal, and exclude the electrons that have undergone dipole scattering to low-angles, which is spatially delocalized.”

3 Reviewer #3 (Remarks to the Author):

The manuscript by Hoglund et al presents a combined experimental and theoretical study of the vibrations at interfaces in oxide perovskite superlattices. With all due respect for the large amount of work involved, and the quite clear presentation, I do not find the study very novel or interesting. The main conclusion, that a small period superlattice simply becomes a homogeneous average structure, is not new, and I would not even call this “emergence”. I agree localized interface states will appear (this is also well known for both electrons and phonons) and will dominate the local and

global behavior for small periods (there is no bulk left), but their properties are not very distinct here, as might be the case, e.g., for 2D electron gases appearing at interfaces of insulators.

We thank Reviewer 3 for taking the time to read and provide a critical review of our manuscript. However, we respectfully disagree regarding the novelty and comments regarding emergence, as detailed in the following discussion. Importantly, we believe the additional experiments outlined below give ubiquitous evidence of ‘emergent’ properties that deviate from that of an ‘average’ structure determined by the constituent layers; the dynamic property evolution that occurs in-tandem with the observed structural evolution has, to the best of our knowledge, not been reported in preceding literature with the resolving capabilities used here. In other words, for the first time, we demonstrate structure-property relationships in superlattices with greatly reduced periodicity that arise *solely* due to interfacial structural coupling/coherence. In our revised manuscript, we consider electronic, optical, and thermal properties of these superlattices as they transition through varying degrees of structural coherence.

First, to determine the underlying electronic properties of these heterostructures, we performed second-harmonic-generation (SHG) experiments. In contrast to linear optical measurements, which are dictated by the average of the linear response of the materials comprising the superlattice (e.g., interface responses are negligible), the higher order optical response tensor associated with SHG vanishes in both STO and CTO bulk layers. In other words, due to a break in inversion symmetry, this technique directly measures the electronic polarizability of bonds comprising the *interfaces* between the layers. As described in our revised manuscript on page 9, these experiments elucidate upon the dynamic evolution of the electronic structure of STO-CTO superlattices, which we find to be directly related to the structural evolution associated with the interfaces. Stated differently, we observe a direct correlation between the electronic polarizability of the STO-CTO structure and the interfacial coupling determined via iDPC-STEM imaging.

While the measured nonlinear optical response is directly coupled to the underlying structure, we note separate consideration of changes in the infrared optical properties due to structural transitions in our investigated superlattices. To this end, we have performed linear dielectric measurements in the infrared, namely, FTIR measurements. For a material system undergoing changes in its vibrational spectra, one would expect the phonon-associated dipole to dictate this frequency range of photons. Due to the reduced thickness of the materials investigated here, a vast fraction of the infrared optical activity is associated with the underlying substrate. Thus, we compare the residual difference between a bare substrate’s FTIR response to that of the superlattice + film heterostructure. As shown on page 4 of our revised manuscript, we observe a consistent change in the IR optical response at $\sim 550\text{ cm}^{-1}$; this response is strongly related to the periodicity of the superlattice. Though the exact mode responsible for this feature is difficult to assign, it demonstrates the relationship between interfacial structural coupling and the optical response of our materials, in, at least qualitative, alignment with the aforementioned nonlinear optical measurements.

Finally, we turn to the thermal response of the structurally-evolving heterostructure. As thermal transport is a direct result of lattice vibrations, one would anticipate a strong correlation between the results demonstrated in our manuscript and the thermophysical properties of the superlattice. Indeed, as described in our prior work, a trend in the thermal conductivity of the SLs that is inconsistent with

incoherent, diffusive phonon propagation is found to exist in these superlattices below a critical periodicity. This anomalous trend in the thermal transport coefficient is ubiquitous evidence of a crossover from incoherent to coherent-like transport of phonons. However, the origin of this crossover had not been elucidated in prior studies – the increase in thermal conductivity associated with coherence is ultimately a direct result of the interfacial structural coupling demonstrated in this work. In other words, the structure-property relationship between coherent phonon transport (and ergo thermal conductivity) remained lacking in oxide superlattices to-date. This work provides direct evidence for interfacial coupling and structural transitions to be the underlying mechanism for such an observation of incoherent-to-coherent transport.

To provide a more direct measure of phonon transport, rather than observations in thermal conductivity, we have performed an additional set of experiments; namely, time-domain Brillouin scattering (TDBS) measurements. These ultrafast pump-probe experiments detailed in our revised manuscript provide a direct measure on the lifetime of longitudinal zone-center phonons in the oxide SL. Previous theory suggests that phonon *group velocities* increase greatly with reduced layer period thickness, due to zone folding, is largely responsible for the thermal conductivity trends in superlattices. However, our experiments verify that the observed trends in thermal conductivity are due to the phonon *scattering times* – these changes in vibrational scattering times are a direct result of the evolving superlattice structure and vibrational spectrum investigated here via high-resolution electron microscopy and spectroscopy. In other words, through a number of experimental approaches, we verify a direct correlation between the vibrational transport properties and the interfacial coupling within the atomistic structure.

In summary, the reviewer's astute comments led us to perform an array of additional experiments to provide a more detailed narrative on the structure-property relationships of the unique structures investigated in our original submission. The results of these measurements, individually detailed in the above paragraphs and in our revised manuscript, strongly support a direct, unprecedented relationship between the observed interfacial coupling regimes of low-periodicity superlattices and the over-arching electronic, optical, and thermal properties of the heterostructure.

These types of superlattices have been synthesized for decades now with atomic layer precision, and the extent of the interface effect (in particular the effect on the oxygen octahedron rotation) has been very widely studied, including by the present authors, both for substrate effects (e.g. JM Rondinelli, NA Spaldin - Physical Review B 82 113402 2010, J He et al. Phys Rev Lett 105 227203 2010) and for interface effects (e.g. May et al Phys Rev B 83 153411 2011, Rispen et al Phys. Rev. B 90, 104106 2014). Claiming that a new symmetry appears at the interface is a bit of a stretch: things stay Pnma with an intermediate angle of rotation (I do understand that some forbidden reflections are lifted). The interface localized vibrational modes mainly interpolate between the bulk frequencies, and the shift is very small. The STEM/EELS identification of the vibrational energies is delicate and impressive, but has been shown before, and is not the claim of novelty of this manuscript.

In summary I find the claims of emergent phenomena and novelty to be a bit inflated, and that the global story is not of the level I would expect for publication in Nature.

We appreciate the reviewer's comments as they have helped us greatly revise our manuscript to increase clarity, without inflation, upon both the novelty and observance of emergent phenomena that are enabled through an understanding of the atomistic structural transitions that arise in these oxide heterostructures. Indeed, the reviewer is certainly correct that oxide superlattices have been grown for decades with atomic-layer precision, and *aspects* of their interfaces have been previously studied. However, to the best of our knowledge, this study is the first demonstration of "local atomic-scale displacements altering the vibrational responses of materials" – the experimental novelty lies within the fact that vibrational electron energy-loss spectroscopy has not been used to connect the local structure of materials to their vibrational responses. As outlined in our above response, the additional experiments obtained following the reviewer's astute comments provide the first direct link of the local structure of materials and their vibrational responses at the atomic level to macroscopic properties ranging from electronic, optical, and thermal transport processes. Not only do these results introduce a novel means of material engineering through interfacial coherence, but they elucidate upon a number of outstanding questions over the last few decades of oxide superlattice growth/characterization, such as phonon coherence, with an atomistic perspective.

The authors would like to elaborate on the comment "vibrational electron energy-loss spectroscopy has not been used to connect the local structure of materials to their vibrational response". All prior vibrational electron energy-loss spectroscopy research focus on the impact of chemical heterogeneity manifesting in localized vibrations. Examples include, the localized vibrations of Si substitutions in Graphene by Hage et al. (2020), delocalized Si-SiO₂ interface vibrational response by Venkatraman et al. (2018) that addresses dipole active Si-O related modes. The lack of structure-vibration relation using vibrational EELS is with the exception of research comparing cubic and hexagonal boron nitride, see Nicholls et al., "Theory of Momentum-Resolved Phonon Spectroscopy in the Electron Microscope." and Hage et al., "Nanoscale Momentum-Resolved Vibrational Spectroscopy.". The two aforementioned works use larger probes to compare experimentally acquired momentum resolved vibrational electron energy-loss spectroscopy to theoretically predicted phonon dispersions of isolated materials. They do not map structural or vibrational changes in a material that result from heterogeneities.

We thank the reviewer for suggesting that the octahedral rotations at the interface do not constitute a new symmetry but are rather an extension of the *Pnma* structure. We have modified our phrasing throughout the manuscript, discussing these rotations as a 'new symmetry associated with local atomic displacements.' At the interfaces the octahedra are not always rigid and can both undergo distortion and rotation. However, we specifically quantify the octahedral *rotation* because it is the main structural parameter that is characterizable within the limits of our resolution and inherent artifacts present in serial image acquisition. The distortions of the octahedra present at the interfaces can result in an entirely new symmetry. Quantifying the distortions would require a precision not achievable with this advanced imaging technique, so we solely focus on the rotations. However, these octahedral rotations directly address concerns over evolving symmetry and whether *Pnma* is a valid description. One must consider that if the magnitude of octahedral tilt is a gradient within the one unit-cell wide structurally diffuse interface, then it *inherently* must break any two-fold symmetries associated with the orthorhombic *Pnma* space group, because the gradient is one-fold. Therefore, the interface must have a new symmetry even if it is derivative of the CTO parent phase. Excitingly, this then raises the interesting question of locally probing the vibrations of a system where the interface structure is not derivative of the bounding materials. For example, a material system that forms an

interface complexion with entirely unique and non-derivative symmetry from either bounding structure, or grain boundaries. However, that is a future direction that is outside the scope of the presented study.

Details:

It is not clear on which basis the SL4 is chosen as intermediate - it looks very similar to SL2 for most purposes. The extent of octahedron rotation influence is not really quantified, but has been examined in past works.

The choice of SL4 as an intermediate layer thickness was chosen because it has a minimum thermal conductivity, see the prior STO-CTO superlattice work Ravichandran et al., "Crossover from Incoherent to Coherent Phonon Scattering in Epitaxial Oxide Superlattices." (2014). The experimentally measured octahedral tilt profiles of SL4 and SL2 are very distinct, as shown in Figure 1(j) compared to Figure 1(m). We have included the Figure and associated caption below for Reviewer 3's convenience. The profiles provided are a direct quantification of the octahedral rotations and agree with the tilt profiles predicted by our DFT calculation, which are shown as dashed curves. The authors additionally provided quantified statistics of the tilt profiles in the supplemental information.

We note that in our additional experiments, outlined above, we considered superlattices spanning a much larger range to further investigate degrees of interfacial coupling on the macroscopic electronic, optical, and thermal properties of the material. All experimental techniques suggest the transition begins at \sim SL4 and the structure has fully evolved by SL2, further validating the choice of SL periodicities for investigation with high resolution electron microscopy.

“Figure 2. *Period dependent changes in the symmetry of STO-CTO superlattices.* (a) Superlattices structures calculated from DFT with schematics of the (left) chemically and (right) structurally defined interfaces. Here green, blue, and cyan rectangles correspond to STO, CTO, and interface layers, respectively; the same colors are used in e,f,h,i,k, and l panels. Green, blue, grey, and red circles in a,e,f,h,i, and k correspond to Sr, Ca, Ti, and O atoms, respectively. [100] zone-axis SADP in (b) SL27, (c) SL4, and (d) SL2 grown on NdGaO₃. Colored arrows correspond to ordered reflections from the three possible domains. Solid arrows indicate ordered reflections that exist, and hollow arrows indicate absences. Insets in show ball-and-stick models of the orientations present with border colors matching the arrows. Red and blue arrows and insets are viewed along an out-of-phase tilt-axis and the yellow are viewed along an in-phase tilt-axis. In (c) and (d) superlattice reflections are seen in the 001 direction. In (b), closely spaced superlattice reflections appear as streaking of the fundamental reflections. (e,h,k) ADF, (f,i,l) iDPC images, and (g,j,m) octahedral tilt-angles of (e-g) SL27, (h-j) SL4, and (k-m) SL2. The legend inside (g) illustrates the (green) in-plane and (black) out-of-plane tilt-angles. The tilt-angles for a one unit-cell column are overlaid in each iDPC image to demonstrate the changing in-plane (green triangles) and out-of-plane (grey triangles) tilt-angles. In (g,j,m), solid and dashed curves are from experimental measurements and calculations, respectively. Error bars represent one standard deviation. Chemically abrupt interfaces are illustrated to the left of ADF images (e,h,k) and model structures (a), illustrating the abrupt change between STO (green) and CTO (blue) layers. Chemically diffuse interfaces are illustrated to the right of iDPC images (f,i,l) and model structures (a), illustrating the non-abrupt symmetry changes that are occurring as a result of octahedral coupling.”

The issue of whether the SL2 angle still oscillates (theory) or not (experiments, though not super clear, there may be one oscillation) is not at all resolved (p.8) in my opinion: could be semi-local DFT is wrong, or missing some Hubbard localization, or experiment not precise enough to distinguish the oscillation left in SL2.

We appreciate the reviewer's critical perspective on this interesting observation. In the series of additional experiments performed, as outlined in our opening response to Reviewer 1, there are a number of complementary measurements suggesting that SL2 lacks oscillations, in contrast to theory yet in agreement with our high-resolution electron microscopy results. In particular, our nonlinear optical SHG results, which are directly sensitive to the interfacial structure of the heterostructure, strongly suggest that SL3 and SL2 lack inversion symmetry, which cannot occur if a gradient or oscillation in the octahedral tilt is present. In regards to the DFT calculations, the choice of exchange-correlation potentials can influence the lattice constants and degree of tilting. Inclusion of a Hubbard would not be appropriate as it typically affects electronic and magnetic properties, not structural and phonon properties. Nevertheless, the agreement between DFT results and experiment for SL4 and SL8 suggests that the predictions for SL2 are reliable for the structures we examined. In Supplemental information Material, we reported calculations that included intermixing in SL2 and the oscillation narrows. Thus, it is very likely that intermixing and/or thermal effects at room temperature may be responsible for the absence of oscillations in the experimental data. We have included the following statement in the Supplemental information Section 3 (page 11) Material:

"The narrowing of the oscillation that is caused by intermixing in Figure S5 is an indication that intermixing and/or thermal effects that can occur at room temperature may be responsible for the absence of oscillations in the experimental data of Figure 1(m)."

For the DFT calculations why was the LDA chosen? It typically underestimates volumes and stabilizes or hardens phonon modes. Is this critical in getting a physical/reasonable phonon spectrum for STO and or CTO?

We appreciate the reviewer's astute questions, as these aspects are key towards the comparison of experiment and theory outlined in our manuscript. We chose the local-density approximation due to a trade-off in computational cost and accuracy. For example, even the LDA, calculations on the SL27 structure are simply infeasible due to cost. However, despite higher efficiency, LDA has proven to be highly accurate for non-chemically-bonded material systems; LDA is generally applicable in systems lacking many-body correlations, which should not play a significant role in either STO or CTO. *These trade-offs associated with LDA are not expected to negatively affect the vibrational spectrum calculations.* In particular, LDA is notably suitable for its ability to reproduce the atomistic structure, including bond lengths and general geometry considerations, while generally having short-comings in its ability to describe electronic band structures and materials with highly-localized orbitals (e.g., transition metals). In short: LDA is well-suited for geometrical and vibrational calculations.

Fig 2a and others, DOS units should be states / energy

Yes, this is very accurate and we thank Reviewer 3 for letting us know that we left the figure axes unlabeled. The authors have provided the correct figures below for Reviewer 3's convenience.

Figure 3. Localized vibrational response of superlattices indicates emergent role of the interfacial symmetry. (a) DFT-calculated phonon DOS projected on the octahedron oxygen and Ti atoms of SL27, SL4, and SL2 models. Arrows indicate the dominant phonon peaks. Cascade of (b) DFT-calculated phonon DOS projected on STO (green), CTO (purple), and interface (orange) layers and the total DOS (black) each for superlattice model. (c-k) Monochromated STEM-EELS line profile analyses of the three SL structures: (c-e) SL27, (f-h) SL4, and (i-k) SL2 each with the (c,f,i) ADF profile, (d,g,j) EELS profile, and (e,h,k) integrated spectra from each layer (as indicated by colored regions in the ADF profile).

Figure S1. A-site projected phonon DOS.

Fig 2k label 1 should be SL2

The authors thank Reviewer 3 for pointing to this typo. We have corrected the legend in Figure 2(k) and have included a copy of the figure below for Reviewer 3's convenience.

Figure 3. Localized vibrational response of superlattices indicates emergent role of the interfacial symmetry. (a) DFT-calculated phonon DOS projected on the octahedron oxygen and Ti atoms of SL27, SL4, and SL2 models. Arrows indicate the dominant phonon peaks. Cascade of (b) DFT-calculated phonon DOS projected on STO (green), CTO (purple), and interface (orange) layers and the total DOS (black) each for superlattice model. (c-k) Monochromated STEM-EELS line profile analyses of the three SL structures: (c-e) SL27, (f-h) SL4, and (i-k) SL2 each with the (c,f,i) ADF profile, (d,g,j) EELS profile, and (e,h,k) integrated spectra from each layer (as indicated by colored regions in the ADF profile).

p.5 "tilts are different"

The authors would like to thank Reviewer 3 for noting this typographical error. We have corrected "different" to "different."

Fig 3c title should be FTIR LSAT

We thank Reviewer 3 for noting the incorrect title so that it can be revised. The Figure has been moved to the Supplemental information as Figure S12 in place of a new Figure. The authors have included the corrected figure below.

Figure S12. Interface driven modifications in the macroscopic Raman and infrared response of STO-CTO superlattices. Raw (solid), fitted (dashed), and residual (dot-dashed) data for (a) UV-Raman and (b) FTIR spectra taken from superlattices on an NGO substrate, and (c) FTIR spectra taken from superlattices on a LSAT substrate. 200 nm STO and CTO thin-films on substrates used to fit superlattice spectra are shown as the bottom of each panel. Difference curves are scaled by a factor of two for clarity.

p.12 "similar in energy to those..."

The authors thank Reviewer 3 for finding the grammatical error. We have changed “similar in energy to *that* observed” to “similar in energy to *those* observed”.

p.15 "rationale for a single phase" then "the layers behave similarly"

The authors thank the Reviewer 3 for letting us know that we refer to SL2 as a single phase but still attribute response to specific layers. The authors agree that this might be confusing. We discussed the decision to assign a STO and CTO layer for vibrational electron energy-loss data in the methods section of our manuscript. We debated on whether or not we should include layers in the SL2 electron energy-loss data, because doing so would be inconsistent with our structural characterization. On the other hand, not including the chemically assigned layers in the SL2 energy-loss data would provide an inconsistent comparison with SL4 and SL27 electron energy-loss data. In the end we decided to include defined layers in the SL2 electron energy-loss data, which then shows that the peak energies are consistent from layer-to-layer emphasizing that the defined layers are responding the same.

p.16 "of a respective constituent layer is averaged per atom" - the whole phrase is a bit convoluted.

The authors have changed “The projected phonon DOS of a respective constituent layers is *averaged* to *per atom*.” to “The projected phonon DOS of a respective constituent layers is *normalized by the*

number of atoms per layer to provide a consistent comparison between the three superlattices.” The new phrasing should better describe the procedure we used to enable comparison between the three superlattice projected phonon density-of-states. The sentence following the sentence in question *“The total phonon DOS of each model is then obtained by $n_{total} = (x \times n_{STO} + y \times n_{CTO} + z \times n_{int.}) / (x + y + z)$, where x, y, and z are the numbers of atoms considered in each layer.”* should also help understand the averaging procedure.

Reviewer Reports on the First Revision:

Referee #1 (Remarks to the Author):

The revised manuscript by Dr Hoglund and co-authors entitled "Nanoscale Phonon Spectroscopy Reveals Emergent Interface Vibrational Structure of Superlattices" has been significantly improved with respect to the initial manuscript. I much appreciate to what depth the authors have considered, discussed and implemented the various comments from the three referees (beyond a close look to the reply of my own remarks, I have also considered with interest the replies to my colleagues).

The new version now exposes in a much clearer way the originality and new gained insight on the important question of how interfaces are constructed in complex perovskite-type heterostructures, how they impact macroscopic physical properties and how different types of heterostructures can allow the tuning of such macroscopic properties. There is no reason that this technique should be limited to perovskites, but its great potential is the likely extension to the wider family of functional materials. Beyond the now clarified originality, the data and experiments are now much better analyzed and have been extended by an array of critical additional experiments. This adds considerable information and supports the various claims. The article is now much better set into the context of relevant literature. All this together now provides a strong and coherent narrative.

In conclusion, I consider that the article is now acceptable for publication in Nature. It is important to realize that new characterization techniques have the potential of strongly influencing a field because they make visible the important materials chemistry and materials physics at play, thus offering the avenue for developing and implementing new concepts, which would otherwise not be thinkable or verifiable. An emblematic example is that nano-physics and technology would likely not have happened without the discovery of atomic force microscopy (AFM). While I am not implying that the here developed technique is comparable to the Nobel-prize winning AFM, I believe that new cutting-edge techniques such as the one discussed in the present article can provide a great impetus for both fundamental understanding, and new device concepts and realizations.

Referee #2:

Recommends publication

Referee #3 (Remarks to the Author):

Manuscript 401706 by Hoglund et al has been revised with new experimental data. As I mentioned before, I am (still) both 1) impressed with the experimental prowess (the use of SHG is a nice addition), and nanoscale diagnostics of structure and dynamics, and 2) disappointed with the "emergent" properties. The authors cite many articles (refs 3,7–9,19,19–24,41) with very interesting emergent physical observables, but here the octahedral rotation angle is predictable and has few physical consequences (variation of the phonon frequency, detectable in the thermal

conductivity). Further if there is intermixing at the interface, there is no emergence, just averaging and alloying for the thinnest layers.

Anyhow, I do not wish to belabor the point with the authors and I grant them their own view. I leave the evaluation of the "sufficient novelty" to the editors. The scientific quality and precision of the manuscript is not in doubt.

Regarding the SHG signal, the inversion symmetry is restored in the geometry for thin heterolayers, but not in the chemical homogeneity - there is something subtle going on with either the similarity of the Sr and Ca responses or the characteristic length scales SHG is actually sensitive to. This should be commented on.

Author Rebuttals to First Revision:

Referee 3 Response

Referee 3 provided one comment to be addressed:

“Regarding the SHG signal, the inversion symmetry is restored in the geometry for thin heterolayers, but not in the chemical homogeneity - there is something subtle going on with either the similarity of the Sr and Ca responses or the characteristic length scales SHG is actually sensitive to. This should be commented on.”

This interesting aspect of our SHG data is one of the unique features associated with the 'emergent' structure observed in this work. As suggested by the reviewer, the lack of an SHG response, despite lack of chemical homogeneity when inversion symmetry is restored for 2x2 and 1x1 superlattices, is likely due to the similarity in Sr and Ca responses and may give insight to the 'local' atomic environment under consideration. Specifically, SHG occurs when an atom experiences anharmonic forces along a given axis; in other words, if a given atom 'senses' asymmetric environments on either side, SHG can be generated. Our results suggest that for thin-enough heterolayers, despite chemical inhomogeneity, a given atom is experiencing symmetric potentials on either side. However, a Sr atom near the interface may have Sr atoms beneath it, and Ca atoms above - clearly an asymmetry. Our results suggest that either the length scale under consideration is much larger than a single atomic plane, or more likely, that the forces imposed by both Sr and Ca on a given atom are highly similar.